# MULTI-SOURCE KNOWLEDGE-FUSION FOR SOURCE-FREE DOMAIN ADAPTATION IN OBJECT DETECTION

## ABSTRACT

Source-free domain adaptation (SFDA) enables adaptation to a target domain without access to source data or labeled target samples, making it particularly valuable in privacy-sensitive applications such as military operations and healthcare. To leverage complementary and transferable knowledge from multiple source domains, multi-source-free domain adaptation (MSFDA) extends SFDA by collectively adapting pre-trained models from multiple sources. However, a key challenge in MSFDA is the significant distribution shift among multiple source and target domains, which often leads to suboptimal performance, especially in complex tasks like object detection. To address this, we propose a novel multi-source knowledge-fusion framework that effectively aggregates knowledge from multiple sources and mitigates distribution discrepancies. We first conduct text-driven feature augmentation that narrows the semantic gap by transforming unlabeled target images into source-stylized images using only textual descriptions of each source domain, such that the pre-trained source models are directly applicable. Each domain expert is then updated with its respective stylized target images, while the aggregator undergoes both local and global updates to ensure stable adaptation. To further improve pseudo-label quality, peer network-based confidence selection is performed to filter out noisy labels. Our method achieves state-of-the-art performance on multiple real-world datasets, demonstrating its effectiveness in multi-source free domain adaptation.

## 1 INTRODUCTION

Source-Free Domain Adaptation (SFDA) has recently gained prominence as a solution to practical challenges such as data privacy, distributed data storage, and inconvenient data transmission Huang et al. (2021); Li et al. (2021b; 2022). SFDA aims to adapt pre-trained model on the source domain to unlabeled target domain without accessing actual source data. Multi-source free domain adaptation extends SFDA to incorporate pre-trained models from multiple source domains, allows knowledge aggregation from a broader variety of data, which is particularly advantageous when the target domain is diverse and spans a wide range of possible scenarios Peng et al. (2019); Dong et al. (2021); Ahmed et al. (2021).

However, properly aggregating knowledge from multiple source domains without access to actual data poses a set of unique challenges. First, the source domains may not comprehensively cover all aspects of the target domain, leading to missing or poorly represented features. Even with multiple source models, certain characteristics of the target domain might remain unaccounted for, hindering generalization and degrading adaptation performance. Second, without access to source data, determining the relative importance of each source model is non-trivial. Some source models may be more relevant than others for a given target domain, but misaligned weighting or reliance on less relevant sources can negatively impact adaptation Li et al. (2024). Last, source models trained on different domains may encode conflicting feature representations, leading to inconsistencies in target predictions Wang et al. (2019); Ding et al. (2016). Disagreements among source models can introduce noise into pseudo-labeling, propagating errors across training stages and ultimately compromising adaptation performance.

While few existing methods have made progress in addressing MSFDA by quantifying the contributions of multiple source models and finding optimal combinations through techniques like joint feature alignment Dong et al. (2021); Ahmed et al. (2021); Peng et al. (2019), or attention mecha-

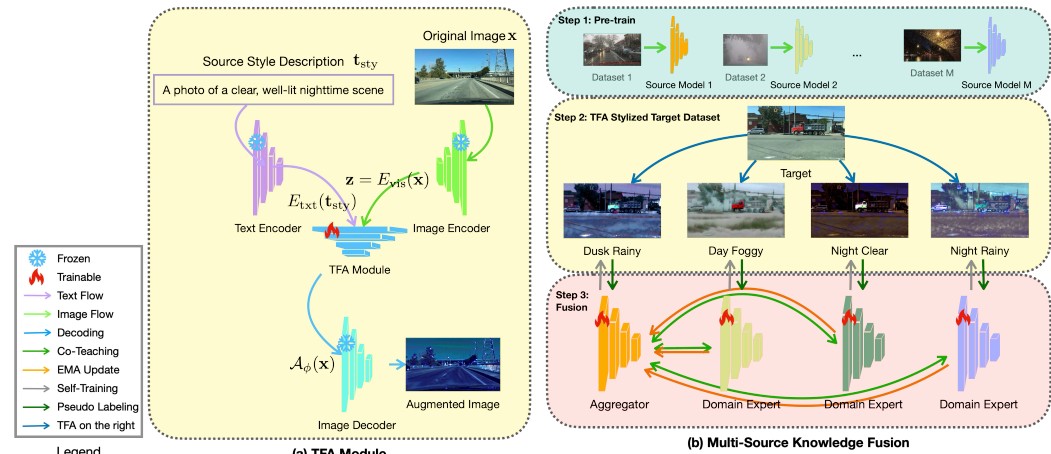

Figure 1: (a) **TFA module:** Given a target image and a style description, TFA uses CLIP features to transform the image representation to match the source domain style. (b) **Overall pipeline:** (1) Pre-train source models on their respective source datasets; (2) Generate stylized target datasets with TFA; (3) Train the multi-source knowledge fusion framework, where domain experts are adapted locally and the aggregator is updated via EMA-based knowledge integration.

nisms Li et al. (2024), these approaches have some inherent limitations. For example, joint feature alignment aims to learn a shared common space from multiple source models for the generalization to target domain by quantifying contribution of each domain or find the optimal combination of source domains Dong et al. (2021); Ahmed et al. (2021); Shen et al. (2023). However, source domains can exhibit significant divergence in feature distributions due to differences in data characteristics, such as lighting conditions, object categories, or scene variations. These differences are often not easily captured through a single alignment process, and aligning features can lead to loss of important domain-specific information. Attention mechanisms have been used to focus on the most relevant parts of the source features during adaptation Li et al. (2024). However, attention alone does not fully address the semantic gaps that arise due to differences in how source models perceive the target data. Attention mechanisms can focus on certain features but may not be able to comprehensively reduce the domain shift between the source and target domains.

To systematically address the challenges outlined above, we propose a novel multi-source knowledge fusion framework, as illustrated in Figure 1, consisting of three key components. The first component conducts Text-driven Feature Augmentation (TFA) to explicitly reduce domain gaps, enabling pre-trained source models to better generalize to unlabeled target images. TFA leverages the vision-language space of foundation models such as CLIP, which aligns image features with textual descriptions Radford et al. (2021). In this space, an image and its corresponding description are positioned closely, allowing text to serve as a proxy for modifying image features. Therefore, we propose TFA to stylize unlabeled target images with source domain characteristics, effectively bridging the semantic gap between pre-trained source models and the target domain without requiring access to original source data.

The second component is a novel multi-source setting designed to ensure stable knowledge integration across heterogeneous source domains. Since source models pre-trained on different domains may encode conflicting feature representations, direct aggregation can cause divergence and hinder adaptation. To address this, we propose a multi-source knowledge fusion framework in which one source model is designated as the aggregator and the others as domain experts. Both the aggregator and domain experts are locally updated through self-training on TFA-stylized images aligned with their respective source domain styles. For cross-domain knowledge integration, the aggregator is globally updated using the EMA of domain experts. To further alleviate domain divergence, we introduce a contribution network that dynamically meta-learns the EMA rate assigned to each domain expert, thereby quantifying its contribution to the aggregator. This network is optimized via an entropy minimization objective, encouraging prediction consistency across source models.

By adaptively weighting domain expert contributions, our approach achieves robust and stable multi-source knowledge fusion.

During local updates, the aggregator and domain experts collaborate to filter out noisy pseudo-labels for each other. In source-free domain adaptation (SFDA), confidence selection is a widely adopted denoising strategy. However, existing methods predominantly rely on self-entropy–based filtering Li et al. (2021b); Shen et al. (2023); Kim et al. (2021), which suffers from a key limitation: as training progresses, models tend to overfit noisy pseudo-labels, becoming increasingly confident in incorrect predictions. This overconfidence amplifies label noise, making it progressively harder to separate clean from noisy samples. To overcome this issue, our third component introduces external validation, enabling the aggregator and domain experts to teach each other rather than relying solely on self-entropy filtering. The aggregator, being more stable, is less affected by specific noisy samples encountered by the domain experts. Conversely, the domain experts provide complementary feedback by identifying mislabeled instances for the aggregator. This mutual refinement process reduces overfitting to noise, stabilizes training, and ultimately improves adaptation performance.

**Summary of contributions.** In this work, we address Multi-Source Free Domain Adaptation (MSFDA) for object detection. Our key contributions are: (1) We propose Text-Driven Feature Augmentation (TFA), which generates source-stylized target images to mitigate domain shifts and improve source model generalization. (2) We develop a multi-source knowledge fusion framework that designates one source model as an aggregator and the others as domain experts. A contribution network adaptively weights expert influence, while a mutual denoising strategy enables them to validate each other's pseudo-labels, reducing noise overfitting and enhancing adaptation stability. (3) We conduct extensive experiments on multiple benchmarks, showing consistent improvements over existing MSFDA methods.

## 2 RELATED WORKS

**Multi-source-free domain adaptation.** Multi-Source-Free Domain Adaptation (MSFDA) aims to distill knowledge from multiple pre-trained models and adapts to an unlabeled target domain without access to the actual source data Dong et al. (2021); Shen et al. (2023); Yeh et al. (2023); Li et al. (2023; 2024); Peng et al. (2019). Dong et al., propose to quantify the contributions of multiple source models with a source-specific transferable perception module. It then improve the quality of the pseudo label with a confident-anchor-induced pseudo label generator Dong et al. (2021). Aiming to find the optimal combination of source models, Ahmed et al., learn a set weights by minimizing the conditional entropy of transferring each source model to the unlabeled target. They also provide intuitive theoretical insights to justify their methodology Ahmed et al. (2021). Shen et al., balance domain aggregation, pseudo-labeling, and joint feature alignment with information-theoretic bound on the generalization error Shen et al. (2023). Focusing on balancing between instance specificity and domain consistency, Li et al., propose a parameter-tuning free method for MSFDA with a attention module that learns both intra-domain weights and inter-domain ensemble weights Li et al. (2024). In contrast to these existing methods, our method focuses on explicitly reducing the semantic gaps among different source models and the unlabeled target data distribution. We propose a text-driven feature augmentation technique to achieve style transfer given only images from the target domain, and a simple description of the source domain style.

**Source-free domain adaptation for object detection.** Source-free domain adaptation for object detection (SFOD) operates under the assumption that only the pre-trained model on the source domain is accessible, while the actual source data is not available, presenting itself as a promising area of research. Conventional SFOD methods commonly employ the pseudo labeling paradigm, involving a cyclic process of model adaptation that oscillates between predicting pseudo labels and fine-tuning the model Huang et al. (2021); Li et al. (2021b); Xiong et al. (2021). Some recent efforts attempt to address these problems by using self-entropy descent as a confidence threshold to select high-quality pseudo labels Li et al. (2021b). Other efforts directly learn domain-invariant features through devising domain perturbation Xiong et al. (2021), graph alignment constraint Li et al. (2022), adversarial alignment of the target images Chu et al. (2023), instance relation graph network Vibashan et al. (2023), or teacher-student models Lin et al. (2023); Liu et al. (2023). Although existing SFOD methods Li et al. (2021b); Lin et al. (2023); Liu et al. (2023); Li et al. (2022) have shown promise,

MSFDA is less explored for object detection tasks. In this work, we propose to utilize multiple pre-trained source models to address source-free domain adaption for object detection.

**Text-based style transfer.** Style transfer aims to transform a content image by transferring the semantic texture of a style image. Traditional style transfer approaches require a reference style image for learning the style texture the texture to change the style of the content image, which might not be always available. Under this condition, using text information to conveys the desired style has emerged as a solution. Current text-based style transfer methods can be categorized into two parts: (1) The generative-based methods; StackGAN Zhang et al. (2017a) integrated text conditions to multi-scale generative model for high-quality image synthesis. AttnGAN Xu et al. (2018) further improved the performance with attention mechanism on text and image features. ManiGAN Li et al. (2020) proposed a modules for simultaneously embedding the text and image features. StyleCLIP Patashnik et al. (2021) performed attribute manipulation with exploring learned latent space of StyleGAN Karras et al. (2019). StyleGAN-NADA Gal et al. (2022) proposed a model modification method with using text conditions only, and modulates the trained model into a novel domain without additional training images. (2) The non-generative-based methods; CLIPstyler Kwon & Ye (2022) design a modulation of the style of content images only with a single text condition using the pre-trained text-image embedding model of CLIP, and propose a patch-wise text-image matching loss with multiview augmentations for realistic texture transfer. PODA Fahes et al. (2023) propose a prompt-driven instance normalization (PIN) layer to do style transfer, where affine transformations of low-level features are optimized such that the representation in CLIP latent space matches the one of text-based prompt. PromptStyler Cho et al. (2023) simulates various distribution shifts in the joint space by synthesizing diverse styles via prompts without using any images to deal with source-free domain generalization. In this work, we propose a novel non-generative text-based style transfer method, TFA, that focuses on aligning the target image with the source style text in both high-level semantics and low-level visual textures.

## 3 METHODOLOGY

The proposed approach aims to reduce the domain shift between multiple source domains and a target domain. To achieve this, we introduce text-driven feature augmentation (TFA), which uses simple textual descriptions of the source domains to augment target images with corresponding styles. This allows for the direct application of pre-trained models to the augmented target images. Additionally, we propose a multi-source knowledge fusion framework to integrate knowledge from multiple source models. To enhance the quality of pseudo-labels, we use confidence selection in a mutual refinement manner, offering a more robust approach to handling noisy labels.

**Problem formulation.** We consider $M$ labeled source domains denoted as $\{D^{s_i}\}_{i=1}^M$, where each domain $D^{s_i} = \{(\mathbf{x}_k^{s_i}, y_k^{s_i})\}_{k=1}^{N_i}$ consists of $N_i$ images. Each image $\mathbf{x}_k^{s_i}$ is paired with an annotation $y_k^{s_i} = (\mathbf{b}_k^{s_i}, c_k^{s_i})$, where $\mathbf{b}_k^{s_i}$ denotes the bounding box coordinates and $c_k^{s_i}$ represents the class label of the object in the $k$-th image of the $i$-th source domain. In addition, we are given an unlabeled target domain $D^t = \{\mathbf{x}_k^t\}_{k=1}^{N_t}$, which contains $N_t$ images without annotations. During adaptation, we do not have access to the raw source data. Instead, we are provided with: (1) the $M$ pre-trained source models $\{\theta^{s_i}\}_{i=1}^M$, each trained on a different source domain, and (2) the text descriptions $\{\mathbf{t}_{\text{sty}}^{s_i}\}_{i=1}^M$ that characterize the style of each source domain. The goal is to leverage the pre-trained source models and their textual style descriptions to adapt effectively to the unlabeled target domain.

**CLIP.** We leverage the vision–language (V-L) space of pre-trained CLIP models Radford et al. (2021) for TFA. CLIP consists of a visual encoder $E_{\text{vis}}$ and a text encoder $E_{\text{txt}}$, which map images and text into a shared embedding space. Given an image $\mathbf{x}$, the visual encoder produces $\mathbf{v} = E_{\text{vis}}(\mathbf{x})$. For the $i$-th category, we construct a prompt "A photo of a [class-$i$]", tokenize it as $\mathbf{p}_i$, and encode it into $\mathbf{t}_i = E_{\text{txt}}(\mathbf{p}_i)$. The prediction for class $y$ is obtained via a softmax over cosine similarities:

$$p(\hat{y} = y|\mathbf{x}) = \frac{\exp(\cos(\mathbf{v}, \mathbf{t}_y)/\tau)}{\sum_{k=1}^K \exp(\cos(\mathbf{v}, \mathbf{t}_k)/\tau)}. \tag{1}$$

This aligns visual and textual features, enabling cross-modal representation learning.

## 3.1 TEXT-DRIVEN FEATURE AUGMENTATION

To bridge the semantic gap between pre-trained source models and target images, we propose Text-Driven Feature Augmentation (TFA), which transforms target image features to match the source domain style guided by textual descriptions. The style transformation is performed by aligning the target image's features with textual representations of the source domain style in a shared vision–language space, where image and text features are well-aligned. By minimizing the gap between the target image feature and the textual source style feature, the target image is effectively transformed to reflect the source domain style. Specifically, we introduce a learnable augmentation module $\mathcal{A}_\phi(\cdot)$ that adapts target features in the vision–language (V-L) space. Given a target image $\mathbf{x}$, its CLIP visual feature is extracted as $\mathbf{z} = E_{\text{vis}}(\mathbf{x})$, and a style description of a source domain is encoded as $E_{\text{txt}}(\mathbf{t}_{\text{sty}})$. The augmentation module generates $\mathcal{A}_\phi(\mathbf{z})$, which is optimized to align with the textual style while preserving image content. The learning objective is:

$$\phi^* = \min_\phi \lambda_1 \mathcal{L}_{\text{style}} + \lambda_2 \mathcal{L}_{\text{content}} + \lambda_3 \mathcal{L}_{\text{Gram}}, \tag{2}$$

$$\mathcal{L}_{\text{style}} = 1 - \cos(\mathcal{A}_\phi(\mathbf{z}), E_{\text{txt}}(\mathbf{t}_{\text{sty}})), \tag{3}$$

$$\mathcal{L}_{\text{content}} = \|\mathcal{A}_\phi(\mathbf{z}) - \mathbf{z}\|_1, \tag{4}$$

$$\mathcal{L}_{\text{Gram}} = \|\text{Gram}(\mathcal{A}_\phi(\mathbf{z})) - E_{\text{txt}}(\mathbf{t}_{\text{sty}})\|_2^2, \tag{5}$$

We implement $\mathcal{A}_\phi$ as a learnable feature augmentation network parameterized by $\phi$. As shown in Figure 1 (a), $\mathcal{A}_\phi$ is a lightweight multilayer perceptron (MLP) that takes the CLIP visual feature $\mathbf{z} = E_{\text{vis}}(\mathbf{x})$ and textual feature $E_{\text{txt}}(\mathbf{t}_{\text{sty}})$ as input and outputs a transformed feature $\mathcal{A}_\phi(\mathbf{z})$ aligned with the source domain style. The parameters $\phi$ are optimized with the objective in Eq. (2), while the CLIP encoders $E_{\text{vis}}$, $E_{\text{txt}}$, and the image decoder remain frozen. Finally, a pretrained ClipStyler image decoder Kwon & Ye (2022) is used to reconstruct stylized images from $\mathcal{A}_\phi(\mathbf{z})$. This design ensures that TFA operates efficiently in the joint V-L feature space without requiring image-space generation.

In Eq. (3), $\cos(\cdot)$ computes the cosine similarity between augmented feature embedding and style text embedding: $\cos(\mathcal{A}_\phi(\mathbf{z}), E_{\text{txt}}(\mathbf{t}_{\text{sty}})) = \frac{\mathcal{A}_\phi(\mathbf{z}) \cdot E_{\text{txt}}(\mathbf{t}_{\text{sty}})}{\|\mathcal{A}_\phi(\mathbf{z})\| \|E_{\text{txt}}(\mathbf{t}_{\text{sty}})\|}$, which measures the semantic alignment between the augmented target image features and the text description of the source domain style. By maximizing the cosine similarity, we ensure that the style of the augmented image is consistent with the desired source style, which is represented in the CLIP text embedding space. Eq. (4) acts as an $\mathcal{L}_1$ regularizer to match high-level image content between the augmented image and the original target image. In Eq. (5), $\text{Gram}(\cdot)$ computes the Gram matrix, capturing style features like texture and lighting distributions. By minimizing the difference between the Gram matrix of the augmented image and the source style, this term encourages the augmented image to adopt the broader source domain style characteristics. Specifically, $\mathcal{L}_{\text{style}}$ focuses on semantic alignment including high-level concepts and object structures, while $\mathcal{L}_{\text{Gram}}$ focuses on texture and appearance similarity including low-level statistics like color and texture patterns. Together, they ensure that the augmented images resemble the source domain both in high-level semantic meaning and low-level visual texture, making them more useful for adapting pre-trained models to the target.

## 3.2 MULTI-SOURCE KNOWLEDGE FUSION FRAMEWORK

Using TFA, we construct multiple stylized target datasets $\{D^{t \to s_1}, D^{t \to s_2}, ..., D^{t \to s_M}\}$, where $D^{t \to s_i} = \{\mathcal{A}_{\phi_i}^{s_i}(\mathbf{x}) | \mathbf{x} \in D^t\}$. In this section, we introduce the multi-source knowledge fusion framework, which effectively integrates knowledge from multiple source models while minimize the negative impact of noisy pseudo labels during adaptation. Specifically, given pre-trained source models $\{\theta^{s_1}, \theta^{s_2}, \ldots, \theta^{s_M}\}$ as domain experts, we designate one model as the aggregator $\theta^{\text{agg}}$, and denote the remaining models as domain experts $\theta_i^{\text{DE}}$. The aggregator simultaneously acts as a domain expert for its own source domain while also serving as the central model for integrating knowledge across all domains. During adaptation, both the aggregator and domain experts are locally updated on their corresponding augmented target datasets, while the aggregator further aggregates cross-domain knowledge through an additional EMA update from the domain experts.

**Domain expert update via self-training.** Each domain expert $\theta_i^{\text{DE}}$ (including the aggregator $\theta^{\text{agg}}$ of its own domain) is adapted to its corresponding augmented dataset $D^{t \to s_i} = \{\mathcal{A}_{\phi_i}^{s_i}(\mathbf{x}) \mid \mathbf{x} \in D^t\}$ via self-training. Specifically, the expert generates pseudo labels $\tilde{y}_i$ for the augmented images

$\mathbf{z}_i = \mathcal{A}_{\phi_i}^{s_i}(\mathbf{x})$, which are then used to update itself with the local detection loss:

$$\theta_i^{\text{DE}} \leftarrow \theta_i^{\text{DE}} - \gamma_i \nabla_{\theta_i^{\text{DE}}} \mathcal{L}_i^{\text{local}}, \tag{6}$$

where $\gamma_i$ is the learning rate. The local loss $\mathcal{L}_i^{\text{local}}$ follows the standard Faster R-CNN detection loss Ren et al. (2015):

$$\mathcal{L}_i^{\text{local}} = \mathcal{L}_{cls}^{rpn}(\mathbf{z}_i, \tilde{y}_i) + \mathcal{L}_{reg}^{rpn}(\mathbf{z}_i, \tilde{y}_i) + \mathcal{L}_{cls}^{roi}(\mathbf{z}_i, \tilde{y}_i) + \mathcal{L}_{reg}^{roi}(\mathbf{z}_i, \tilde{y}_i). \tag{7}$$

Here, $\mathcal{L}_{cls}^{rpn}$ and $\mathcal{L}_{reg}^{rpn}$ are the classification and regression losses of the region proposal network (RPN), while $\mathcal{L}_{cls}^{roi}$ and $\mathcal{L}_{reg}^{roi}$ are those of the region-of-interest (ROI) head. The classification terms are standard cross-entropy losses for category prediction, and the regression terms measure localization error. In this way, each domain expert iteratively improves itself by using its own high-confidence predictions as supervision, following the principle of self-training.

**Aggregation via meta-learning.** To integrate knowledge across domains, the aggregator $\theta^{\text{agg}}$ is updated as an exponential moving average (EMA) of the domain experts' parameters:

$$\theta^{\text{agg}} \leftarrow \alpha^{\text{agg}} \theta^{\text{agg}} + \sum_{i=1}^{M-1} \alpha_i^{\text{DE}} \theta_i^{\text{DE}}, \tag{8}$$

where $\alpha^{\text{agg}}$ and $\{\alpha_i^{\text{DE}}\}_{i=1}^{M-1}$ are EMA rates that control how much each model contributes to the aggregator. Instead of using fixed weights, we let the contributions $\boldsymbol{\alpha} = [\alpha^{\text{agg}}, \alpha_1^{\text{DE}}, ..., \alpha_{M-1}^{\text{DE}}]$ be dynamically learned by a small meta-network $\mathcal{F}(\cdot)$. The intuition is that not all domain experts are equally reliable—some may provide more consistent or less noisy knowledge than others. To train $\mathcal{F}$, we encourage agreement among models by minimizing the entropy of their averaged predictions on target images:

$$\boldsymbol{\alpha}^* = \min_{\boldsymbol{\alpha}} - \sum_{k=1}^{K} \tilde{p}(y_k|\mathbf{x}) \log \tilde{p}(y_k|\mathbf{x}), \text{ where } \tilde{p}(y_k|\mathbf{x}) = \frac{1}{M} \sum_{i=1}^{M} p(y_k|\mathbf{x}, \theta^{s_i}), \tag{9}$$

where $p(y_k|\mathbf{x}, \theta^{s_i})$ is the class probability vector produced by model $\theta^{s_i}$. For notation clarity, we use $\{\theta^{s_i}\}_{i=1}^{M}$ instead of $\theta^{\text{agg}}$ and $\{\theta_i^{\text{DE}}\}_{i=1}^{M-1}$. In practice, $\theta^{\text{agg}}$ is updated by Eq. (8), and $\boldsymbol{\alpha}$ is produced by $\mathcal{F}(\cdot)$. Since $\mathcal{F}$ is trained to reduce prediction entropy, it learns to assign higher weights to models that produce confident and consistent predictions on target data, while down-weighting noisier experts.

Eq. (9) defines a bi-level meta-optimization, in which the EMA rate $\alpha$ acts as a meta-parameter optimized over the performance of the aggregator $\theta^{agg}$ on the target domain. This process is mathematically well-defined and follows standard meta-learning optimization principles. The update in Eq. (9) corresponds to a meta-gradient $\nabla L(\theta^{agg}(\alpha))$ where $\theta^{agg}(\alpha)$ (i.e., $\theta^{agg}$) is the result of the inner-level update in Eq. (8). This is a gradient-through-gradient computation related bi-level optimization frameworks. Therefore, the optimization dynamics directly follow from well-established meta-learning theory. The inner (Eq. (8)) and outer optimization (Eq.(9)) ensures that $\alpha$ learns the weight of each source model should contribute to the aggregator (including the aggregator) through EMA so that the aggregated representation remains stable across different sources. The reviewer's concern about undefined optimization dynamics is addressed by (a) using SGD for the meta-update and (b) applying only a single meta-step per batch, which ensures stability and avoids overfitting. The dual-level design, where EMA is treated as a learnable meta-parameter controlling aggregation strength across sources, is fundamentally different from prior works that use static EMA schedules. This mechanism is essential to achieving robust multi-source domain adaptation, and Eq. (9) provides the formal learning rule that makes this adaptation possible.

**Self-training with mutual confidence selection.** Confidence selection is usually coupled with self-training to filter out noisy pseudo labels in SFDA. Usually, it filters out less confident predictions with high entropy via a pre-defined threshold Li et al. (2021b); Kim et al. (2021). This approach suffers from a critical limitation: as training progresses, the model tends to overfit noisy pseudo-labels, becoming increasingly confident in incorrect predictions. This overfitting amplifies label noise, making it progressively harder to distinguish clean from noisy samples. To address this issue, we introduce an external validation mechanism by enabling the aggregator and domain experts to teach each other Han et al. (2018).

For example, given a domain expert $\theta_i^{\text{DE}}$ and its corresponding augmented target dataset $D^{t \to s_i}$, the aggregator $\theta^{\text{agg}}$ computes the loss of each example and rank them in ascending order, where examples with loss higher than a predefined threshold will be considered as noisy labels and be discarded. The rationale behind this mechanism is twofold: (1) The aggregator and domain experts are trained with different augmentations of the same target data distribution. This corelation allows each model to identify and filter out noisy labels that might have been incorrectly assigned in the early stages of training. (2) If the domain expert were used for confidence selection, errors from noisy pseudo-labels generated during early training would propagate and accumulate, leading to the model becoming increasingly overfitted to these erroneous labels. As the model continues to train, it would gradually lose the ability to distinguish between correct and noisy labels. However, by using the aggregator, which is less affected by overfitting to the domain expert's noisy data, we can mitigate this issue and ensure that the pseudo-labels used for domain expert training remain more accurate and reliable. Similarly, we randomly choose a domain expert to filter out the noisy pseudo labels of the aggregator.

## 4 EXPERIMENTS

In this section, we first introduce the datasets, evaluation metrics, comparative baselines and the implementation details. Then the quantitative and qualitative results are presented to prove the effectiveness of our method. Some additional results are presented in the Appendix.

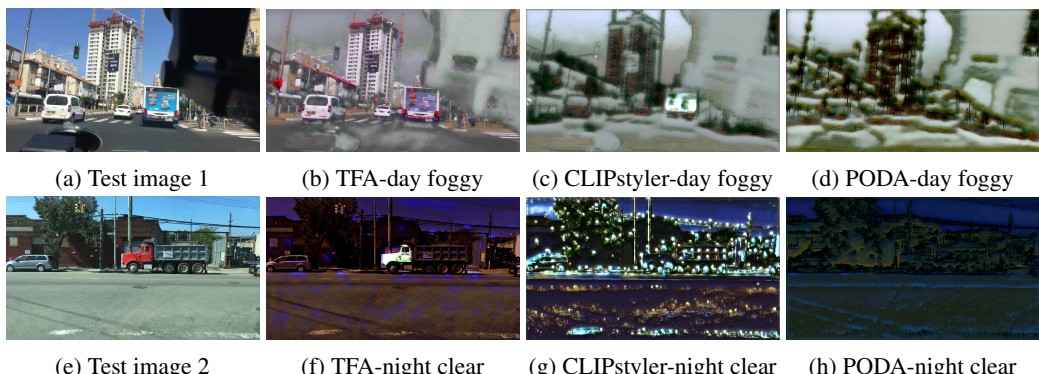

| (a) Test image 1 | (b) TFA-day foggy | (c) CLIPstyler-day foggy | (d) PODA-day foggy |
| (e) Test image 2 | (f) TFA-night clear | (g) CLIPstyler-night clear | (h) PODA-night clear |

Figure 2: Two images (a)/(e) from Day Clear are stylized with different domain styles: Day Foggy and Night Clear, using different text-based style transfer methods: TFA, CLIPstyler and PODA.

### 4.1 EXPERIMENT DETAILS

**Datasets and evaluation metric.** For evaluation, we adopt three cross-domain datasets: (1) Diverse Weather Dataset (DWD) Wu & Deng (2022), which includes driving scenes under various weather and time conditions; (2) Cityscapes Cordts et al. (2016), FoggyCityscapes Sakaridis et al. (2018), and KITTI Geiger et al. (2013a), a combination that covers both real-world urban driving scenes and synthetic car images; and (3) Art Inoue et al. (2018), which consists of images rendered in diverse artistic styles. More detailed dataset descriptions are provided in the Appendix. In all our experiments, we use the Mean Average Precision (mAP) as our metric. Specifically, we report the mAP@0.5, which considers a prediction as a true positive if it matches the ground-truth label and has an intersection over union (IOU) score of more than 0.5 with the ground-truth bounding box. All reported experimental results are based on a sinle run.

**Comparison baselines** We compare our method with (1) *Source-free domain adaptation (SFDA)* methods including SED Li et al. (2021b), HCL Huang et al. (2021), and IRG Vibashan et al. (2023), LODS Li et al. (2022), LPLD Yoon et al. (2024), SF-UT Hao et al. (2024), DRU Khanh et al. (2024), and (2) *Multi-source-free domain adaptation (MSFDA)* methods including Mean-Teacher Tarvainen & Valpola (2017), MixUp Zhang et al. (2017b), MSFDAOD Zhao et al. (2024), CAiDA Dong et al. (2021), Selective Self-Training (SST) Shen et al. (2023), Bi-ATEN Li et al. (2024). For those SFDA and MSFDA methods that are not designed for object detection task, the reported results are based on our re-implementation. We additionally report the performance of Faster R-CNN (FR) Ren et al. (2015) initialized with ImageNet pre-trained weights.

**Implementation details.** For pre-trained source models, we used the implementation of Faster R-CNN Ren et al. (2015) from the MMDetection library Chen et al. (2019). We use ResNet50 as the backbone, with the learning rate equal to 0.01 and the max epoch set to 8. And we use the last checkpoint as our source model. The training is conducted using 4 P4. The network architecture for $\mathcal{F}(\cdot)$ is a 3×3 convolutions and 64 filters, followed by batch normalization, a ReLU nonlinearity, and 2×2 max-pooling. For TFA, we use the Layer 1 target feature maps of the pre-trained CLIP-ResNet-50 model Radford et al. (2021). To optimize the augmentations with $\mathbf{t}_{sty}$, we generate random crops from the source images and re-size them to $320 \times 320$ pixels. The style parameters $\phi$ are 256D real vectors. The CLIP embeddings are 1024D vectors. We use ChatGPT to generate the description of $\mathbf{t}_{sty}$ (in less than 10 words) with the template prompt "a photo of" and the dataset description of DWD (given by Wu & Deng (2022)). The resulting text description of each source domain are shown in Table 6. Optimization was done for 100 iterations using SGD with a learning rate of 1, momentum of 0.9, and weight decay of 1e-4. The training is conducted using 1 A100. During adaptation, we first initialize the target model with the chosen closest pre-trained source model. Then with the backbone frozen, we fine-tune the classifier and bounding box head with the augmented feature and pseudo labels. Non-maximum Suppression (NMS) Hosang et al. (2017) is utilized to eliminate duplicate detections and select the most relevant bounding boxes that correspond to the detected objects. The learning rate is 0.01, and the max epoch is 4. The training is conducted using 2 A100.

## 4.2 MAIN RESULTS

Table 1 presents a comprehensive comparison between our method and state-of-the-art SFDA and MSFDA approaches for object detection on the DWD dataset, which includes five domains: Day Clear (DC), Night Clear (NC), Day Foggy (DF), Dusk Rainy (DR) and Night Rainy (NR). We first observe that MSFDA methods consistently outperform SFDA approaches, reaffirming the advantage of leveraging diverse domain knowledge. The presence of multiple source domains provides a richer representation space, enabling better generalization to the target domain. Secondly, our proposed method outperforms all existing MSFDA methods across all four domains. This demonstrates the effectiveness of our approach in mitigating domain gaps by incorporating both local and global updates in a structured multi-source knowledge fusion paradigm. Additionally, by learning domain-specific contributions dynamically, the multi-source knowledge fusion framework avoids the pitfalls of naïve model aggregation, leading to more stable adaptation. A detailed per-class analysis for each domain is provided in the Appendix.

Table 1: Multi-source domain adaptation results (mAP). For each target domain, Day Clear and the rest three domains are used as the source domains for the multi-source setting. For the single-source UDA and SFDA, Day Clear is used as the source following the typical setting Wu & Deng (2022); Vidit et al. (2023); Fahes et al. (2023).

| Method | Multi-Source | mAP | | | |
|---|---|---|---|---|---|
| | | NC | DR | NR | DF |
| FR | ✗ | 34.4 | 26.0 | 12.4 | 32.0 |
| SED | ✗ | 33.4 | 21.1 | 15.1 | 29.4 |
| HCL | ✗ | 33.8 | 21.9 | 16.3 | 30.2 |
| IRG | ✗ | 42.7 | 30.5 | 18.4 | 35.2 |
| LODS | ✗ | 33.5 | 25.7 | 13.5 | 31.2 |
| LPLD | ✗ | 34.7 | 28.5 | 14.2 | 32.8 |
| SF-UT | ✗ | 36.8 | 30.0 | 16.9 | 34.2 |
| DRU | ✗ | 35.7 | 28.5 | 15.8 | 33.4 |
| Mean-Teacher | ✓ | 44.1 | 32.0 | 19.1 | 36.8 |
| MixUp | ✓ | 36.0 | 30.0 | 16.7 | 31.5 |
| MSFDAOD | ✓ | 42.1 | 30.8 | 18.7 | 34.4 |
| CAiDA | ✓ | 43.4 | 31.7 | 19.5 | 35.2 |
| SST | ✓ | 43.8 | 32.0 | 19.6 | 35.4 |
| ATEN | ✓ | 43.9 | 32.2 | 19.8 | 35.5 |
| Bi-ATEN | ✓ | 43.9 | 32.3 | 19.5 | 35.4 |
| **Ours** | ✓ | **44.5** | **32.5** | **20.3** | **38.4** |

**Additional results on the Art dataset** The results in Table 2 show consistent improvements of our method across all three domains when tested under the "leave-one-domain-out" setting. Compared with prior approaches such as Mean-Teacher, MixUp, and MSFDAOD, our method achieves the highest performance on Clipart1k (48.45%), Watercolor2k (50.33%), and Comic2k (46.39%). The gains over the strongest baseline Bi-ATEN are 1.62%, 1.57%, and 1.15% respectively. These results highlight that our method is not only effective in the weather/lighting scenarios but also generalizes well to domain shifts caused by stylistic variations, such as differences between clipart, watercolor, and comic images. The improvements across all target domains indicate that the combination of domain-specific self-training and adaptive aggregation contributes to robust cross-domain transfer even under large appearance gaps.

**Per-class analysis.** In Table 3, we present a per-class analysis on the Foggy-Cityscapes dataset using Cityscapes and KITTI as source domains. The results demonstrate that our method achieves the highest mAP by consistently ranking at the top across multiple classes. Similarly, MSFDA methods outperform SFDA methods, highlighting the advantages of leveraging multiple source domains.

Table 2: Comparison of detection performance (mAP %) on the Art dataset across three domains: Clipart1k, Watercolor2k, and Comic2k.

| Method | Clipart1k | Watercolor2k | Comic2k |
|---|---|---|---|
| Mean-Teacher | 41.98 | 42.55 | 40.32 |
| MixUp | 41.26 | 41.98 | 39.85 |
| MSFDAOD | 42.43 | 44.29 | 41.37 |
| CAiDA | 43.99 | 45.42 | 42.87 |
| SST | 46.45 | 48.33 | 44.92 |
| Bi-ATEN | 46.83 | 48.76 | 45.24 |
| **Ours** | **48.45** | **50.33** | **46.39** |

Table 3: Per-class results on Foggy-Cityscapes

| Method | Multi-Source | Bus | Bike | Car | Motor | Person | Rider | Truck | Train | mAP All |
|---|---|---|---|---|---|---|---|---|---|---|
| SED | ✗ | 11.8 | 34.3 | 40.4 | 34.5 | 21.7 | 44.0 | 32.6 | 25.3 | 30.6 |
| SED(Moisac) | ✗ | 22.2 | 39.0 | 40.7 | 34.1 | 25.5 | 44.5 | 33.2 | 28.4 | 33.5 |
| HCL | ✗ | 25.0 | **46.0** | 41.3 | **35.9** | 26.9 | 40.7 | 33.0 | 28.1 | 34.6 |
| SOAP | ✗ | 37.2 | 37.9 | 48.4 | 31.8 | 35.9 | 45.0 | 23.9 | 24.3 | 35.5 |
| LODS | ✗ | 39.7 | 37.8 | 48.8 | 33.2 | 34.0 | **45.7** | 27.3 | 19.6 | 35.8 |
| IRG | ✗ | 39.6 | 41.6 | **51.9** | 31.5 | 37.4 | 45.2 | 24.4 | 25.2 | 37.1 |
| LPLD | ✗ | 37.4 | 37.8 | 48.7 | 32.0 | 36.1 | 45.3 | 24.0 | 24.5 | 35.6 |
| SF-UT | ✗ | 38.5 | 40.5 | 50.7 | 30.8 | 36.5 | 44.6 | 23.5 | 24.1 | 36.1 |
| DRU | ✗ | 39.2 | 40.9 | 50.9 | 31.2 | 36.7 | 44.9 | 23.9 | 24.3 | 36.4 |
| Mean-Teacher | ✓ | 39.2 | 40.2 | 47.0 | 27.6 | 35.9 | 45.0 | 31.2 | 27.1 | 36.3 |
| MixUp | ✓ | 38.5 | 39.8 | 44.0 | 26.1 | 30.4 | 42.8 | 29.1 | 26.4 | 34.2 |
| MSFDAOD | ✓ | 39.9 | 33.2 | 47.3 | 29.5 | 33.8 | 45.0 | 32.4 | 29.8 | 37.6 |
| CAiDA | ✓ | 41.8 | 44.5 | 47.3 | 29.8 | 34.0 | 45.8 | 33.5 | 32.0 | 37.8 |
| SST | ✓ | 42.2 | 44.2 | 44.9 | 30.3 | 33.5 | 47.2 | 36.0 | 33.2 | 37.9 |
| ATEN | ✓ | 42.6 | 44.4 | 48.8 | 30.0 | 34.1 | **47.5** | 36.4 | **33.3** | 38.1 |
| Bi-ATEN | ✓ | **42.6** | 44.5 | 47.9 | 29.7 | 34.0 | 47.4 | **36.5** | 33.1 | 38.0 |
| **Ours** | ✓ | 40.1 | 39.8 | 49.9 | 35.4 | 40.4 | 46.1 | 35.5 | 27.5 | **39.2** |

## 4.3 ABLATION STUDY

**Effectiveness of different components**   In this ablative study, we investigate the impact of each individual component in our framework. The detailed setting and results are shown in Table 4. We observe that, compared to a single source, the utilization of the multi-source knowledge fusion framework largely improves the mAP by 2.1 and the improvement is observed over all categories. By using the TFA, we managed to minimize the domain gap between all the source models and target data, which increased the mAP from 33.5 to 34.2. The confidence selection further improve the performance by 4.2 by filtering out noisy labels.

**Text-driven feature augmentation**   In Figure 2, we show some stylized image rendered with TFA and the other two text-based style transfer methods. Take two images from the Day Clear domain, we augmented them with Day Foggy and Night Clear styles using TFA, CLIPstyler Kwon & Ye (2022) and PODA Fahes et al. (2023), respectively. From the augmented images, we observe that TFA renders augmented images with desirable style effects, including low-level texture and high-level concepts (like fog) while keeping the original content from being distorted. However, for CLIPstyler and PODA, the rendered style effects shroud the original content, making the objects in the original images hard to recognize. We present more qualitative results for TFA in the Appendix.

## 4.4 QUALITATIVE STUDY

We show some qualitative performance in Figure 3 with an example from Night Clear target domain. We observe that SS falsely detects multiple cars as one, misclassifies the Bus as Car, and fails to detect Person. By using the multi-source knowledge fusion, less mistakes were made, showing that the utilization of multiple sources help improve the generalization to some extend. While couple with TFA, MS+TFA successfully detects multiple cars, since the application of TFA reduce the domain gap between the multiple source model and the target images. Along with confidence selection, the noisy labels are filtered out, and we are able to detect all the objects.

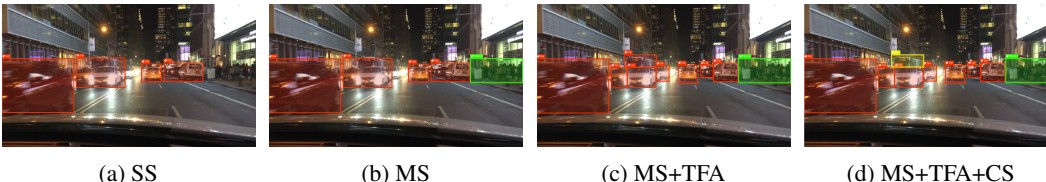

| (a) SS | (b) MS | (c) MS+TFA | (d) MS+TFA+CS |

Figure 3: Qualitative visualization: Bounding box predictions from different settings, where SS stands for single source model, MS stands for multi-source knowledge fusion, and CS stands for confidence selection.

Table 4: Per-class results on Day Foggy with different components enabled. SS stands for single-source model, MS stands for multi-source knowledge fusion, and CS stands for confidence selection. As usual, for single-source, Day Clear is used as the source model, and for multi-source, Day Clear and the rest three targets except for Day Foggy are used as source models. The last row is our proposed method.

| Source | SS | MS | TFA | CS | Bus | Bike | Car | Motor | Person | Rider | Truck | mAP All |
|---|---|---|---|---|---|---|---|---|---|---|---|---|
| ✗ | ✓ | ✗ | ✗ | ✗ | 30.8 | 29.3 | 28.5 | 32.7 | 30.8 | 32.4 | 35.8 | 31.4 |
| ✗ | ✗ | ✓ | ✗ | ✗ | 38.5 | 30.6 | 29.9 | 35.5 | 34.1 | 33.9 | 38.0 | 33.5 |
| ✗ | ✗ | ✓ | ✓ | ✗ | **38.8** | 31.2 | 30.5 | **36.4** | 34.8 | 34.5 | **38.4** | 34.2 |
| ✗ | ✗ | ✓ | ✓ | ✓ | 37.4 | **36.5** | **45.8** | 35.9 | **40.8** | **36.0** | 36.0 | **38.4** |

**Additional Results.** We present additional experiment results in the Appendix, including per-class analysis for each domains in DWD dataset; ablation study of using different source domain text descriptions; impact of the hyperparameter $\alpha$, and impact of heterogeneous source model architectures; impact of the choice of aggregator; ablation on computational cost; additional qualitative results for stylized target images; and comparison with unsupervised domain adaptation methods, and so on.

## 5  CONCLUSION

In this study, we introduce a novel approach to perform multi-source-free domain adaptation, addressing the challenge of integrating multiple sources to harness information effectively. Our method first mitigates domain shift between multiple source domains and the target domain by transforming target images to match the styles of the source domains, utilizing text-based style transfer with textual descriptions of the source domains. Furthermore, we aggregate the information from multiple source models with a novel knowledge fusion framework, where the aggregator and domain experts are updated globally and locally, and the pseudo label quality are mutually enhanced. Experimental evaluations on diverse weather datasets demonstrate the efficacy of our proposed model across different domains. In future research, we plan to explore dynamic target domains where the target data distribution evolves over time.

## 6  REPRODUCIBILITY STATEMENT

We have taken multiple steps to ensure the reproducibility of our work. A detailed description of our proposed method and training objectives is provided in Section 3 of the main paper. Additional pseudo code of the proposed algorithm, detailed training steps, implementation details, hyperparameter settings, and dataset information are included in Appendix B. To further facilitate reproducibility, we provide an anonymous link to the source code and scripts for training and evaluation in Appendix E. All datasets used in our experiments are publicly available, and their references are properly provided.

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

# Supplementary Material

# Appendix

## Table of Contents

# A NOTATIONS AND ALGORITHMS

In this section, we present the table of notations used in the main paper, and the pseudo code for our algorithms. Table 5 summarizes the major notations used in the paper. Our algorithm consists of two main components. In Algorithm 1, we introduce the pseudo code for TFA. In Algorithm 2, we outline the multi-source free domain adaptation with the mean-teacher framework and the confidence selection via co-teaching.

Table 5: Notations

| Notation | Description |
|---|---|
| $D^{s_i}$ | $i$-th source domain |
| $D^t$ | target domain |
| $\theta^{s_i}$ | pre-trained $i$-th source domain model |
| $x_k^{s_i}$ | $k$-th feature from $i$-th source domain |
| $y_k^{s_i}$ | $k$-th label from $i$-th source domain |
| $b_k^{s_i}$ | $k$-th bounding box from $i$-th source domain |
| $c_k^{s_i}$ | $k$-th class label from $i$-th source domain |
| $x_k^t$ | $k$-th feature from target domain |
| $M$ | number of source models |
| $N_i$ | number of data samples in $i$-th source domain |
| $N_t$ | number of data samples in target domain |
| $\phi$ | TFA augmentation parameters |
| $E_{\text{txt}}$ | CLIP text encoder |
| $E_{\text{vis}}$ | CLIP image encoder |
| $\cos(\cdot)$ | cosine similarity |
| $\mathbf{t}_{\text{sty}}$ | domain style description |
| $\mathcal{L}_{\text{style}}$ | text image style consistent loss |
| $\mathcal{L}_{\text{content}}$ | content preservation loss |
| $\mathcal{L}_{\text{Gram}}$ | low-level Gram regularization loss |
| $D^{t \to s_i}$ | augmented target domain datasets with $i$-th source domain style |
| $\theta^{\text{agg}}$ | aggregator |
| $\theta_i^{\text{DE}}$ | $i$-th domain expert |
| $\mathcal{L}_i^{\text{local}}$ | $i$-th local loss |
| $\mathcal{L}_{cls}^{rpn}$ | region classification loss |
| $\mathcal{L}_{reg}^{rpn}$ | region proposal loss |
| $\mathcal{L}_{cls}^{roi}$ | bounding box classification loss |
| $\mathcal{L}_{reg}^{roi}$ | bounding box regression loss |
| $\boldsymbol{\alpha}$ | EMA learning rates |

---

**Algorithm 1** Text-driven Feature Augmentation (TFA)

---

1: **INPUT** Target dataset $D^t$, text description of style $T^{src} = \{\mathbf{t}_{\text{sty}}^{s_i}\}_{i=1}^M$ of source domain $s_i$, $\phi$
2: **OUTPUT** Multiple augmented datasets $\{D^{t \to s_i}\}_{i=1}^M$
3: **for** each target image $\mathbf{x} \in D^t$ **do**
4:     Using CLIP image encoder to extract target image feature: $\mathbf{z} = E_{\text{vis}}(\mathbf{x})$
5:     **for** each source style text $\mathbf{t}_{\text{sty}}^{s_i} \in T^{src}$ **do**
6:         Using CLIP text encoder to extract source text feature: $E_{\text{txt}}(\mathbf{t}_{\text{sty}}^{s_i})$
7:         **while** not converged **do**
8:             Obtain augmented image feature with $s$-th source style using $\mathcal{A}_\phi(\cdot)$: $\mathcal{A}_\phi(\mathbf{z})$ (i.e. $E_{\text{vis}}(\mathcal{A}_\phi(\mathbf{x}))$)
9:             Update $\phi$ with Equation (2)
10:         **end while**
11:     **end for**
12: **end for**

---

---

**Algorithm 2** Multi-Source Domain Adaptation (MSDA)

---

1: **INPUT** Multiple pre-trained source models $\{\theta^{s_i}\}_{i=1}^{M}$, target dataset $D^t$, and multiple stylized target datasets $\{D^{t \to s_i}\}_{i=1}^{M}$, EMA parameters $\boldsymbol{\alpha}$, target dataset $D^t$
2: Choose a pre-trained source model as aggregator $\theta^{\mathrm{agg}}$
3: Choose the rest pre-trained source models as domain expert $\{\theta_1^{\mathrm{DE}}, ..., \theta_{M-1}^{\mathrm{stu}}\}$
4: **while** not converged **do**
5:    // Update aggregator and domain experts
6:    **for** each model $\theta^{s_i}$ in $\{\theta^{s_i}\}_{i=1}^{M}$ **do**
7:      Sample a batch of data from $D^{t \to s_i}$
8:      Generate pseudo label $\tilde{y}_i$ with $\theta^{s_i}$
9:      // Confidence selection in a co-teaching manner
10:      **if** the model is a domain expert **then**
11:        Use the aggregator to filter out noisy labels
12:      **else**
13:        Randomly select a domain expert to filter out noisy labels
14:      **end if**
15:      Update $\theta^{s_i}$ with Equation (7)
16:    **end for**
17:    Update aggregator with Equation (8)
18:    // Update EMA parameters $\boldsymbol{\alpha}$
19:    Sample a batch of data from target dataset $D^t$
20:    **for** each model $\theta^{s_i}$ in $\{\theta^{s_i}\}_{i=1}^{M}$ **do**
21:      Compute the prediction probability of each model on the data
22:    **end for**
23:    Update $\boldsymbol{\alpha}$ with Equation (9)
24: **end while**

---

## B EXPERIMENTAL DETAILS

### B.1 IMPLEMENTATION DETAILS

**Implementation details of $\mathcal{F}$.**   $\mathcal{F}$ is a lightweight neural network designed for meta-learning the EMA parameters. It consists of a $2 \times 2$ followed by two $3 \times 3$ convolutional layers, each with ReLU activations and channel dimensions of 16, 32, and 64, respectively. A final $1 \times 1$ convolutional layer produces a vector of unnormalized scores $\mathbf{s} \in \mathbb{R}^M$, where $M$ is the number of models (aggregator + domain experts). These scores are normalized via a softmax function, which ensuring that all $\alpha_i \geq 0$ and $\sum_{i=1}^{M} \alpha_i = 1$. For training , we use Adam with an initial learning rate of $5 \times 10^{-4}$. Training is conducted over 200 iterations, and the learning rate is reduced by half at iteration 100.

**Implementation details of the multi-source knowledge fusion framework.**   To complement the pseudo-code, we outline the training and evaluation procedure of our proposed method:

- **Step 1: Pretraining source models.** We first pretrain a set of source models on their respective labeled source-domain datasets. The backbone architecture of each model follows the experimental setting (e.g., Faster R-CNN, ATSS, or YOLOv7).

- **Step 2: Pretraining the feature augmentation module.** The feature augmentation module is trained following the procedure in Algorithm 1.

- **Step 3: Generating target-augmented datasets.** Using the trained augmentation module, we translate target-domain samples into multiple source-domain styles, producing augmented datasets for subsequent training.

- **Step 4: Training the knowledge fusion framework.** With both pretrained source models and target-augmented datasets, we train the proposed multi-source fusion framework as described in Section 3.2 and detailed in Algorithm 2. In this stage, domain experts are updated via self-training on the target-style data, while the aggregator progressively integrates knowledge across domains through meta-learned EMA updates.

Table 6: LLM generated source style description for DWD.

| Domain | Style Descriptions |
|---|---|
| Day Clear | "A photo of a clear, sunny daytime scene." |
| Night Clear | "A photo of a clear, well-lit nighttime scene." |
| Night Rainy | "A photo of a rainy, dimly-lit nighttime scene." |
| Dusk Rainy | "A photo of a rainy scene at dusk." |
| Day Foggy | "A photo of a foggy scene during the day." |

Table 7: LLM generated source style description for Art.

| Domain | Style Descriptions |
|---|---|
| Clipart | "An illustration in a flat, clipart style." |
| Comic | "A drawing in bold, comic-book style." |
| Watercolor | "A painting in soft, watercolor brush strokes." |

- **Step 5: Evaluation.** After training, the aggregated model is evaluated on the held-out target domain. We report standard detection metrics (e.g., mAP, mAP@0.5) to assess the performance and cross-domain generalization ability of the framework.

## B.2 DETAILS OF THE DATASETS

The images of the DWD dataset was selected from three primary datasets, Berkeley Deep Drive 100K (BBD-100K) Yu et al. (2020), Cityscapes Cordts et al. (2016) and Adverse-Weather Hassaballah et al. (2020). Additionally, rainy images are rendered by Wu et al. (2021), and some of the foggy images are synthetically generated from Sakaridis et al. (2018). The daytime clear dataset consists of 27708 images, the night clear dataset contains 26158 images, the dusk rainy dataset has 3501 images, the night rainy dataset has 2494 images, and the daytime foggy dataset has 3775 images. All the datasets contain bounding box annotations for the 7 classes objects: *bus, bike, car, motorbike, person, rider*, and *truck*. For text augmentation, we utilize the domain description in Table 6.

For the combined dataset, **Cityscapes**[1] Cordts et al. (2016), **Foggy-Cityscapes**[2] Sakaridis et al. (2018), and **KITTI**[3] Geiger et al. (2013b) for further evaluation. Cityscapes consist of 2975 training images and 500 testing images, have a total of 8 categories captured under normal weather. Foggy-Cityscapes applys images of Cityscapes to simulate foggy as well as inherits the annotations of Cityscapes. KITTI contains 7,481 urban images of the same classes which are different from Cityscapes. For the comparative baselines, the training set of Cityscapes are utlized to pre-trained the source model, and test on the test set of Foggy-Cityscapes following the general setting Xu et al. (2020); Li et al. (2021a). In addition, we incorporate the validation set of KITTI dataset as an additional source model, which includes 1870 images, and then test our model on the test set of Foggy-Cityscapes. For text augmentation, we utilize the domain description in Table 12.

The Art dataset contains different artistic styles including Clipart1k, Comic2k, and Watercolor2k Inoue et al. (2018). Clipart1k contains 1000 clipart images across 20 classes, Watercolor2k and Comic2k contains 2000 watercolor/comic images across 6 classes. For text augmentation, we utilize the domain description in Table 7.

## C ADDITIONAL EXPERIMENTAL RESULTS

In this section, we first provide the per-class analysis of each domain of DWD dataset in Appendix C.1. We than provide experimental results on the Art dataset in Table 2. In Appendix C.2, we conduct ablation studies including the impact of source domain style descriptions, the impact of TFA hyperparameters $\lambda_1$, $\lambda_2$ and $\lambda_3$, the impact of heterogeneous architecture of source models, the impact of

---

[1]https://github.com/tiancity-NJU/da-faster-rcnn-PyTorch
[2]https://www.cityscapes-dataset.com/downloads/
[3]http://www.cvlibs.net/datasets/kitti/

aggregator, and some qualitative results. In Appendix C.3, we show some qualitative results of using TFA with different source descriptions, and their detection results. In Appendix C.5, we compare the qualitative results of TFA with other two text-based style transfer methods CLIPstyler Kwon & Ye (2022) and PODA Fahes et al. (2023).

## C.1 Per-class Analysis of the DWD Dataset

In this section, we present per-class analysis for each domain of the DWD Dataset, including Day Foggy, Dusk Rainy, Night Clear and Night Rainy.

**Multi-source→Day Foggy.** We show the detailed results of each class of Day Foggy in Table 8. In general MSFDA outperform SFDA in most categories, except for Car, FR ranks the top. Our method achieves the best performance on Bus and Truck, and performs consistently better on other classes, resulting the top mAP performance.

**Multi-source→Dusk Rainy.** In Table 9, our method outperforms SFDA and MSFDA in most of the categories except for Car, Motor and Truck, achieving the best average mAP over all categories. Compared to SFDA, our method either ranks the top or the second except for Motor, resulting in the top 1 average mAP over all categories. Mean-Teacher ranks the top for Car and Motor, but our method excel it by outperforming on other classes consistently.

**Multi-source→Night Clear.** In Table 10, our method performs consistently better in most categories, ranking the top at four of them, achieving the highest mAP. For SFDA, FR ranks the top in Car. And for MSFDA, Mean-Teacher ranks the top at Bus and Motor.

**Multi-source→Night Rainy.** As shown in Table 11, in the more challenging Night Rainy dataset, our method achieve the best average mAP by performing consistently well through in two categories, and consistently better in other categories.

Table 8: Per-class results on multi-source to Day Foggy (the setting of source models is the same as in Table 1).

| Method | Multi-Source | AP | | | | | | | mAP |
| --- | --- | --- | --- | --- | --- | --- | --- | --- | --- |
| | | Bus | Bike | Car | Motor | Person | Rider | Truck | All |
| FR | ✗ | 28.1 | 29.7 | **49.7** | 26.3 | 33.2 | 35.5 | 21.5 | 32.0 |
| SED | ✗ | 28.4 | 29.1 | 28.5 | 24.1 | 33.9 | 30.4 | 32.7 | 29.4 |
| HCL | ✗ | 32.5 | 31.3 | 32.1 | 25.9 | 28.0 | 34.2 | 31.8 | 30.2 |
| IRG | ✗ | 33.8 | 33.9 | 34.2 | 36.8 | 37.5 | 38.9 | 34.8 | 35.2 |
| LODS | ✗ | 28.5 | 29.4 | 33.8 | 29.7 | 34.5 | 34.9 | 21.2 | 31.2 |
| LPLD | ✗ | 28.9 | 30.2 | 34.2 | 30.5 | 34.8 | 35.8 | 23.0 | 32.8 |
| SF-UT | ✗ | 30.0 | 30.8 | 35.4 | 31.7 | 35.8 | 36.9 | 24.2 | 34.2 |
| DRU | ✗ | 29.8 | 30.2 | 35.0 | 31.0 | 35.2 | 36.1 | 22.8 | 33.4 |
| Mean-Teacher | ✓ | 35.4 | **37.9** | 40.2 | **39.2** | 31.5 | 33.4 | 32.9 | 36.8 |
| MixUp | ✓ | 33.2 | 32.4 | 33.5 | 26.8 | 29.1 | 35.5 | 33.2 | 31.5 |
| MSFDAOD | ✓ | 31.5 | 32.8 | 36.0 | 33.4 | 38.2 | 38.9 | 29.5 | 34.4 |
| CAiDA | ✓ | 32.4 | 33.6 | 37.2 | 34.2 | 39.5 | 39.9 | 30.4 | 35.2 |
| SST | ✓ | 32.8 | 33.9 | 37.4 | 34.5 | 40.3 | **40.8** | 30.5 | 35.4 |
| ATEN | ✓ | 33.0 | 34.1 | 37.9 | 34.8 | **40.6** | 41.1 | 30.9 | 35.5 |
| Bi-ATEN | ✓ | 32.7 | 34.2 | 37.5 | 34.2 | 40.4 | 40.7 | 30.6 | 35.4 |
| **Ours** | ✓ | **36.4** | 35.5 | 45.8 | 34.9 | 39.8 | 35.0 | **35.0** | **38.4** |

## C.2 Additional Ablation Studies

In this section, we present the ablation study of using different source domain text descriptions, the impact of TFA hyperparameters $\lambda_1$, $\lambda_2$, and $\lambda_3$ in Equation (2), and the impact of heterogeneous source model architectures.

Table 9: Per-class results on multi-source to Dusk Rainy (the setting of source models is the same as in Table 1).

| Method | Multi-Source | AP | | | | | | | mAP |
| | | Bus | Bike | Car | Motor | Person | Rider | Truck | All |
|---|---|---|---|---|---|---|---|---|---|
| FR | ✗ | 28.5 | 20.3 | **58.2** | 6.5 | 23.4 | 11.3 | 33.9 | 26.0 |
| SED | ✗ | 20.4 | 21.5 | 20.6 | 20.8 | 27.5 | 18.3 | 24.5 | 21.1 |
| HCL | ✗ | 26.7 | 14.2 | 22.4 | 14.2 | 22.9 | 14.3 | 30.5 | 21.9 |
| IRG | ✗ | 35.2 | 21.4 | 29.8 | 15.9 | 26.5 | 22.4 | 38.7 | 30.5 |
| LODS | ✗ | 27.1 | 30.9 | 23.4 | 19.7 | 16.3 | 30.7 | 28.2 | 25.7 |
| LPLD | ✗ | 27.9 | 31.4 | 24.2 | 22.5 | 19.9 | 31.5 | 29.2 | 28.5 |
| SF-UT | ✗ | 28.2 | 31.6 | 24.5 | 22.9 | 20.9 | 31.8 | 29.4 | 30.0 |
| DRU | ✗ | 28.0 | 31.2 | 24.5 | 22.4 | 20.4 | 31.6 | 29.4 | 28.5 |
| Mean-Teacher | ✓ | 29.0 | 31.2 | 33.8 | **35.6** | 30.5 | 27.8 | 26.2 | 32.0 |
| MixUp | ✓ | 30.5 | 30.1 | 28.8 | 34.7 | **32.5** | 28.4 | 28.2 | 30.8 |
| MSFDAOD | ✓ | 30.4 | 30.3 | 29.2 | 34.9 | 32.3 | 28.1 | 28.7 | 30.8 |
| CAiDA | ✓ | 31.2 | 31.1 | 30.5 | 35.4 | 33.5 | 28.9 | 29.7 | 31.7 |
| SST | ✓ | 31.4 | 31.3 | 30.8 | 35.5 | 33.8 | 29.1 | 30.2 | 32.0 |
| ATEN | ✓ | 31.5 | 31.4 | 31.1 | 35.8 | 34.0 | 29.0 | 30.4 | 32.2 |
| Bi-ATEN | ✓ | 31.6 | 31.2 | 31.5 | 35.9 | 34.2 | 29.3 | 30.4 | 32.3 |
| **Ours** | ✓ | **34.5** | **33.1** | 30.4 | 29.5 | 31.8 | **34.6** | **34.7** | **32.5** |

Table 10: Per-class results on multi-source to Night Clear (the setting of source models is the same as in Table 1).

| Method | Multi-Source | AP | | | | | | | mAP |
| | | Bus | Bike | Car | Motor | Person | Rider | Truck | All |
|---|---|---|---|---|---|---|---|---|---|
| FR | ✗ | 34.7 | 32.0 | **56.6** | 13.6 | 37.4 | 27.6 | 38.6 | 34.4 |
| SED | ✗ | 31.9 | 34.5 | 33.8 | 31.2 | 32.5 | 34.9 | 33.7 | 33.4 |
| HCL | ✗ | 33.4 | 32.9 | 33.4 | 33.1 | 34.7 | 35.1 | 34.5 | 33.8 |
| IRG | ✗ | 43.2 | 41.8 | 42.4 | 42.4 | 43.5 | 44.1 | 42.4 | 42.7 |
| LODS | ✗ | 32.0 | 34.6 | 33.9 | 31.4 | 32.4 | 35.1 | 33.8 | 33.5 |
| LPLD | ✗ | 33.2 | 32.7 | 33.5 | 33.2 | 34.5 | 35.0 | 34.6 | 34.7 |
| SF-UT | ✗ | 35.1 | 34.2 | 35.9 | 35.3 | 35.8 | 36.2 | 35.9 | 36.8 |
| DRU | ✗ | 34.0 | 33.5 | 34.8 | 34.8 | 34.4 | 35.8 | 33.9 | 35.7 |
| Mean-Teacher | ✓ | **44.5** | 42.3 | 45.6 | **45.2** | 43.2 | 44.5 | 43.4 | 44.1 |
| MixUp | ✓ | 37.2 | 35.4 | 35.8 | 35.6 | 36.4 | 36.8 | 35.6 | 36.0 |
| MSFDAOD | ✓ | 42.8 | 41.2 | 41.5 | 41.6 | 43.0 | 43.2 | 41.5 | 42.1 |
| CAiDA | ✓ | 43.3 | 41.4 | 41.8 | 41.9 | 43.5 | 43.6 | 41.9 | 43.4 |
| SST | ✓ | 43.9 | 41.8 | 42.4 | 42.6 | 44.0 | 44.1 | 42.4 | 43.8 |
| ATEN | ✓ | 43.9 | 41.9 | 42.6 | 42.8 | 44.1 | 44.3 | 42.5 | 43.9 |
| Bi-ATEN | ✓ | 43.8 | 41.9 | 42.6 | 43.0 | 44.2 | 44.2 | 42.3 | 43.9 |
| **Ours** | ✓ | 44.0 | **42.9** | 43.5 | 44.8 | **44.9** | **45.5** | **45.0** | **44.5** |

**Different source domain text descriptions.** In this ablation, we evaluate the utilization of different source domain text descriptions, which are presented in Table 13. We generate Text Descriptions 1 and 2, which are both relevant to the source domain styles while Text Descriptions 3 are randomly generated prompts given by ChatGPT. As depicted in Figure 4, Text Descriptions 1 and 2 exhibit comparable performance, whereas Text Description 3 significantly underperforms. This is attributed to the substantial disparity between the described styles and the actual source domain styles, thereby introducing domain gaps to the source models and the generated images, which poses challenges for the source model in classification.

**TFA hyperparameter $\lambda_1$, $\lambda_2$, $\lambda_3$.** In this ablation, we test the hyperparameters $\lambda_1$, $\lambda_2$, and $\lambda_3$ for TFA in Eq. (2). In the paper, the default weights of $\lambda_1$, $\lambda_2$, and $\lambda_3$ are 10, 5, 1e-3, respectively.

Table 11: Per-class results on multi-source to Night Rainy (the setting of source models is the same as in Table 1).

| Method | Multi-Source | AP | | | | | | | mAP |
|---|---|---|---|---|---|---|---|---|---|
| | | Bus | Bike | Car | Motor | Person | Rider | Truck | All |
| FR | ✗ | 16.8 | 6.9 | **26.3** | 0.6 | 11.6 | 9.4 | 15.4 | 12.4 |
| SED | ✗ | 15.8 | 14.5 | 14.2 | 18.6 | 6.9 | 16.5 | 18.8 | 15.1 |
| HCL | ✗ | 16.6 | 13.8 | 14.5 | 17.0 | 17.1 | 15.6 | 16.2 | 16.3 |
| IRG | ✗ | 18.5 | 24.8 | 15.4 | 18.2 | 18.3 | 16.7 | 16.4 | 18.4 |
| LODS | ✗ | 16.7 | 7.2 | 25.4 | 0.9 | 12.4 | 10.8 | 16.5 | 13.5 |
| LPLD | ✗ | 16.9 | 9.8 | 25.5 | 3.2 | 15.2 | 12.4 | 18.4 | 14.2 |
| SF-UT | ✗ | 17.4 | 10.5 | 25.2 | 5.2 | 16.5 | 12.8 | 18.2 | 16.9 |
| DRU | ✗ | 17.1 | 10.0 | 25.3 | 3.4 | 15.5 | 12.8 | 18.6 | 15.8 |
| Mean-Teacher | ✓ | 14.9 | 18.8 | 19.4 | 17.6 | 17.2 | 25.5 | **20.7** | 19.1 |
| MixUp | ✓ | 16.9 | 14.2 | 14.8 | 17.5 | 17.7 | 16.0 | 16.5 | 16.7 |
| MSFDAOD | ✓ | 19.2 | 16.4 | 17.0 | 19.1 | 18.0 | 17.2 | 17.4 | 18.7 |
| CAiDA | ✓ | 19.9 | 17.4 | 18.2 | 20.0 | 19.2 | 18.3 | 18.5 | 19.5 |
| SST | ✓ | 19.9 | 17.7 | 18.0 | 19.8 | 19.1 | 18.0 | 18.6 | 19.6 |
| ATEN | ✓ | **20.2** | 18.0 | 18.4 | **20.2** | **19.3** | 18.2 | 18.5 | 19.8 |
| Bi-ATEN | ✓ | 19.8 | 17.5 | 17.8 | 19.6 | 18.8 | 17.8 | 18.5 | 19.5 |
| **Ours** | ✓ | 16.5 | **19.6** | 20.2 | 19.4 | 18.4 | **26.2** | 20.5 | **20.3** |

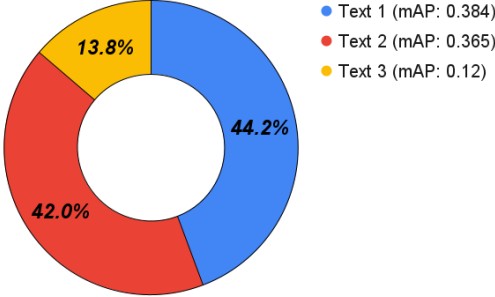

Figure 4: Source Domain Text Description Ablative on Multi-source → Day Foggy: Text 1, 2 and 3 are detailed in Table 13.

Table 12: Source Domain Text Description

| Source Domain | Descriptions |
|---|---|
| Cityscapes | ["Driving at city streets"] |
| KITTI | ["Driving at urban streets"] |

Table 13: Source domain text descriptions

| Source Domain | Text Descriptions 1 | Text Descriptions 2 | Text Descriptions 3 |
|---|---|---|---|
| Day Clear | ["Driving at clear day time"] | ["Sunny day street view"] | ["Lost in enchanted forest"] |
| Night Clear | ["Driving at clear night time"] | ["Clear night time street views"] | ["Time traveler meets past self."] |
| Night Rainy | ["Driving at rainy night time"] | ["Rainy night street view"] | ["Robot falls in love"] |
| Dusk Rainy | ["Driving at rainy dusk"] | ["Rainy dusk street view"] | ["Magic potion gone wrong"] |
| Day Foggy | ["Driving at foggy day time"] | ["Foggy day street view"] | ["Alien invasion thwarted heroically."] |

Table 14: Per-class results on multi-source to Day Foggy with Ablations on TFA hyperparameters in Equation (2).

| Hyperparameters | AP | | | | | | | mAP |
|---|---|---|---|---|---|---|---|---|
| $(\lambda_1, \lambda_2, \lambda_3)$ | Bus | Bike | Car | Motor | Person | Rider | Truck | All |
| $(0.1, 1, 1e-3)$ | 31.0 | 29.6 | 28.4 | 32.8 | 30.8 | 32.1 | 35.6 | 34.3 |
| $(1, 1, 1e-3)$ | 31.4 | 29.8 | 29.1 | 33.4 | 31.5 | 32.8 | 36.2 | 34.9 |
| $(10, 1, 1e-3)$ | 32.1 | 30.7 | 29.6 | 33.8 | 32.0 | 33.1 | **36.5** | 35.5 |
| $(10, 5, 1e-3)$ (default) | **36.4** | **35.5** | **45.8** | **34.9** | **39.8** | **35.0** | 35.0 | **38.4** |

In this ablation, we show some different combination of the weights and the per-class results are shown in Table 14. As observed, by gradually increasing the weight of $\mathcal{L}_{\text{style}}$ from 0.1 to 10, the per-class performances improve steadily, showing the effectiveness of style transfer via text. When we increase the weight of $\mathcal{L}_{\text{content}}$ from 1 to 5, the performance improve from 35.5 to 38.4, showing that preserving the content information is very important during style transfer. The weight of $\mathcal{L}_{\text{Gram}}$ is kept at a small value so that $\mathcal{L}_{\text{style}}$ will play the major role during style transfer.

**Heterogeneous source model architectures.** In our experiment, we use faster R-CNN as both pre-trained source and target models to demonstrate the effectiveness of our proposed multi-source free domain adaptation algorithm. To further validate the effectiveness, we include the additional architecture experiments with ATSS and YOLO-v7 as model architecture choices, and compare with the latest source-free baselines. The results are presented in Table 15. Despite the choices of model architectures, our methods show consistently better performance. The algorithm can be further extended to multiple architecture setting with minor modifications. Specifically, instead of having only one aggregator, we can initialize multiple aggregator-domain expert pairs with the different architectures, where each pair share the same architecture for one dataset. For the EMA update of the aggregator, we update it with the corresponding domain expert, while the domain expert update remains unchanged. That is, the update of the domain expert $\theta_i^{\text{DE}}$ is the same as in Eq. 7. As shown in Eq. 8, the aggregator are updated with the corresponding domain expert with the same architecture. During inference, the ensemble of domain expert is utilized. We present the result of this extended setting as follows, where we set four model pairs, two with faster-RCNN for Night Clear and Dusk Rainy datasets and two ATSS and YOLO-v7 for Night Rainy and Day Foggy datasets, respectively, denoted as FR+ATSS and YOLO-v7 in the table below.

**Impact of the choice of aggregator.** The aggregator is used directly for inference on target domain images without requiring augmentation during testing. Its selection is guided by semantic descriptions of the source domains, using criteria such as weather conditions (e.g., Rainy vs. Clear) or time (e.g., Night vs. Day) to determine the closest match. For example, if the target domain is Night Rainy, either Night Clear or Dusk Rainy can serve as the aggregator because: (1) Both are semantically close to the target domain. (2) The EMA update integrates characteristics of both Night and Rainy, enhancing generalization to Night-Rainy. This selection process ensures the aggregator is optimally suited for inference in the target domain. The results in Table 16 show that, based on semantic

Table 15: Multi-source domain adaptation results (mAP) using different model architectures (ATSS, YOLO-v7). For each target domain, Day Clear and the rest three other domains are used as the source domains for the multi-source setting.

| Method | Multi-Source | Night Clear | Dusk Rainy | Night Rainy | Day Foggy |
|---|---|---|---|---|---|
| SED (ATSS) | ✗ | 32.9 | 20.5 | 24.7 | 28.9 |
| Mean-Teacher (ATSS) | ✓ | 43.2 | 31.4 | 28.6 | 36.1 |
| MixUp (ATSS) | ✓ | 41.5 | 30.4 | 26.1 | 31.0 |
| Ours (ATSS) | ✓ | **44.1** | **32.0** | **28.7** | **37.8** |
| Ours (FR+ATSS) | ✓ | 43.8 | 31.6 | 28.3 | 37.4 |
| SED (YOLO-v7) | ✗ | 34.2 | 22.0 | 26.2 | 30.6 |
| Mean-Teacher (YOLO-v7) | ✓ | 45.2 | 32.8 | 30.0 | 38.1 |
| MixUp (YOLO-v7) | ✓ | 43.0 | 31.7 | 27.5 | 32.8 |
| Ours (YOLO-v7) | ✗ | **45.6** | **33.5** | **30.2** | **39.6** |
| Ours (FR+YOLO-v7) | ✓ | 45.3 | 33.3 | 30.0 | 39.2 |

closeness, Dusk Rainy is the most effective aggregator for Night Rainy, followed by Night Clear as the second-best choice.

Table 16: mAP on Night Rainy using different aggregator.

| Aggregator | Night Clear | Day Foggy | Dusk Rainy | Day Clear |
|---|---|---|---|---|
| mAP | 18.9 | 17.5 | **20.3** | 18.4 |

**Ablation on backbone architectures.** To verify the robustness and generalizability of our proposed method beyond the initial Faster R-CNN + ResNet-50 setup, we evaluate our framework using a more modern detector, DETR with a ViT backbone Carion et al. (2020). This ablation addresses two key questions: (1) whether the proposed framework improvements are architecture-agnostic, and (2) whether the method provides consistent gains even when applied to stronger baselines. The results in Table 17 show that our method consistently outperforms all baselines across every target domain with DETR + ViT. These findings indicate that the method's effectiveness is not tied to a specific backbone; it adapts well to both CNN-based and transformer-based detectors. Moreover, the improvements demonstrate that multi-source knowledge fusion scales with stronger architectures, highlighting its flexibility and applicability across different detection paradigms. This confirms that the framework can robustly handle domain shifts while leveraging the representational power of modern detection backbones.

Table 17: Comparison of methods on the NC → DR → NR → NC adaptation setting.

| Method | NC | DR | NR | NC (cycle) |
|---|---|---|---|---|
| SED | 32.4 | 29.0 | 16.3 | 30.2 |
| Mean-Teacher | 43.1 | 33.0 | 21.4 | 37.2 |
| MixUp | 39.6 | 30.0 | 18.3 | 33.3 |
| Ours | **44.8** | **34.3** | **22.6** | **38.7** |

**Computational costs.** We compare the training and inference efficiency of our method with baseline methods, including MSFDAOD, CAiDA, and Mean Teacher, in terms of inference speed (FPS) and model size (millions of parameters). As shown in Table 18, our method achieves competitive inference speed while maintaining a relatively compact model size compared to other multi-source adaptation approaches. These results highlight the practicality and deployability of our framework in real-world scenarios.

**Impact of different text-based style transfer techniques.** In this ablation, we investigate the effect of different text-based style transfer methods, including our proposed TFA, PODA Fahes et al. (2023), and ClipStyler Kwon & Ye (2022). As noted earlier, TFA achieves superior style transfer by effectively altering the image style while preserving content integrity. We further evaluate the performance

Table 18: Comparison of inference speed (FPS) and model size (millions of parameters) for different methods.

| Method | Inference Time (FPS) | Parameter Size (M) |
|--------|---------------------|--------------------|
| MSFDAOD | 17 | 133 |
| CAiDA | 15 | 180 |
| Mean-Teacher | 17 | 128 |
| Ours | 17 | 126 |

of our multi-source knowledge fusion framework using target images augmented by PODA and ClipStyler on the Day Foggy adaptation. With PODA-augmented images, the framework achieves 37.5, while ClipStyler achieves 38.2, both of which are lower than TFA's 38.4. This demonstrates that the choice of style transfer technique has a measurable impact on adaptation performance, and that TFA provides the most effective domain-stylized augmentations for our framework.

**Impact of fixed EMA rate.**    In this ablation, we investigate the effect of using fixed EMA rates instead of dynamically learned ones. Specifically, for domain experts (excluding the aggregator), we assign equal weights $\alpha^{\mathrm{DE}} = \frac{1-\alpha^{\mathrm{agg}}}{M-1}$ and vary the aggregator's EMA rate $\alpha^{\mathrm{agg}}$ from 0.50 to 0.99. The resulting mAP values are:

| $\alpha^{\mathrm{agg}}$ | 0.50 | 0.80 | 0.95 | 0.99 |
|------|------|------|------|------|
| mAP | 37.5 | 37.7 | 38.2 | 38.1 |

As shown, although performance slightly improves with higher $\alpha^{\mathrm{agg}}$, the best result (38.2 mAP) still falls short of our full method with adaptive weighting (38.4 mAP). This demonstrates that simply assigning equal importance to non-aggregator models is suboptimal, and that our proposed meta-learned weighting strategy meaningfully contributes to the final performance.

**Evaluation on segmentation and classification tasks.**    Our framework is task agnostic, which can also be applied to classification and segmentation tasks. To further support the generality of the framework, we additionally conduct experiments on classification (DomainNet Leventidis et al. (2021), Table 19) and semantic segmentation (ACDC Sakaridis et al. (2021), Table 20), confirming that the same framework transfers well to other tasks without architectural changes.

| **Domain** | AP | $\mathrm{AP_{base}}$ | $\mathrm{AP_{novel}}$ |
|--------|------|------|------|
| Clipart | 88.65 | 90.87 | 87.24 |
| Infograph | 86.84 | 87.57 | 86.82 |
| Painting | 84.27 | 85.53 | 84.75 |
| Quickdraw | 85.43 | 84.96 | 86.44 |
| Real | 83.62 | 84.58 | 82.66 |
| Sketch | 87.34 | 87.52 | 86.43 |

Table 19: Performance across DomainNet domains.

| **Domain** | AP | $\mathrm{AP_{base}}$ | $\mathrm{AP_{novel}}$ |
|--------|------|------|------|
| Foggy | 32.5 | 33.4 | 29.8 |
| Nighttime | 32.2 | 32.8 | 29.7 |
| Rainy | 29.3 | 30.7 | 26.5 |
| Snowy | 29.1 | 30.2 | 26.6 |

Table 20: Performance across BDD100K weather conditions.

### C.3    QUALITATIVE RESULTS

From Figure 5-Figure 9, we show some qualitative results by making use of the pre-trained image decoder from CLIPstyler Kwon & Ye (2022) of some stylized target images generated with TFA by

Table 21: Text Descriptions

| Source Domain | Descriptions |
|---|---|
| Day Clear | ["Driving at clear day time"] |
| Night Clear | ["Driving at clear night time"] |
| Night Rainy | ["Driving at rainy night time"] |
| Dusk Rainy | ["Driving at rainy dusk time"] |
| Day Foggy | ["Driving at foggy day time"] |
| City streets | ["Driving at city streets"] |
| Urban street | ["Driving at urban streets"] |
| Video game | ["Driving in a video game"] |
| Random Text 1 | ["Robot falls in love"] |
| Random Text 2 | ["Lost in enchanted forest"] |

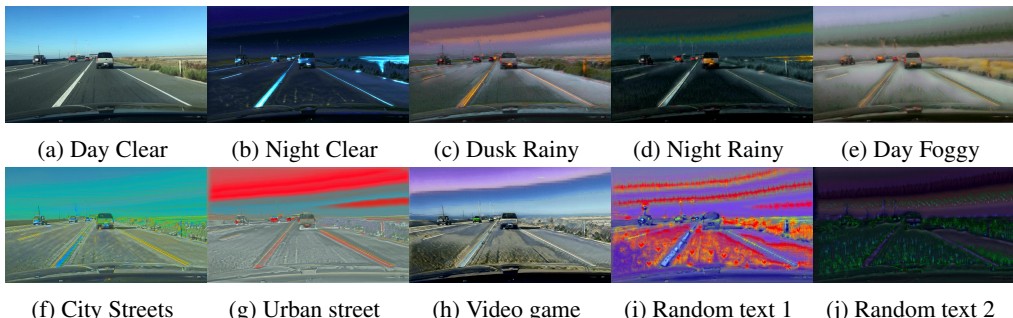

(a) Day Clear  (b) Night Clear  (c) Dusk Rainy  (d) Night Rainy  (e) Day Foggy

(f) City Streets  (g) Urban street  (h) Video game  (i) Random text 1  (j) Random text 2

Figure 5: Stylized images of different styles with image sampled from Day Clear domain of DWD dataset. (a) is the original target image, and (b)-(j) are text-based stylized images with corresponding description in Table 21.

sampling sampled from different domains with different text-based description styles described in Table 21. In general, the text-based style transfer successfully transfer the domain style from the source domain to the target domain with the text descriptions. For example, in Figure 5, taking an image from Day Clear domain, the Night Clear and Night Rainy styles transfer the image to a dark night environment; the Dusk Rainy style imparts a pink dusk ambiance to the image; the Day Foggy style introduces fog into the image. In conclusion, the text-based style transfer technique is able to change the weather and time conditions given an image. In other instances, the City View and Urban View styles largely maintain their similarity in style descriptions, as they are closely related. Conversely, the Video Game style transforms a realistic image into a simulated one. Additionally, the random texts effectively incorporate relevant elements corresponding to the text descriptions.

In addition, we present some qualitative results on the evaluation of the aforementioned augmented images in Figure 10, with the target image sampled from Day Clear and the augmented images from Figure 5. As observed in the first row, when directly predicting the original image with different pre-trained source models, the domain gap between them tends to lead to the erroneous predictions. While in the second row, the pre-trained source models perform well on the corresponding augmented stylized images even they are not sampled from the corresponding domains. They all give perfect predictions except for Night Rainy, which has the largest domain gap with Day Clear. This show that the text based style transfer has reasonably reduced the domain gap between the source models and the target image.

### C.4 ILLUSTRATION OF FAILED EXAMPLES

In this section, we present some examples where the model struggles to detect objects in the target images, as shown in Figure 11. The results show that under extreme conditions, where the domain gap is large, or the classes are unseen in the source, the model's performance significantly degrades.

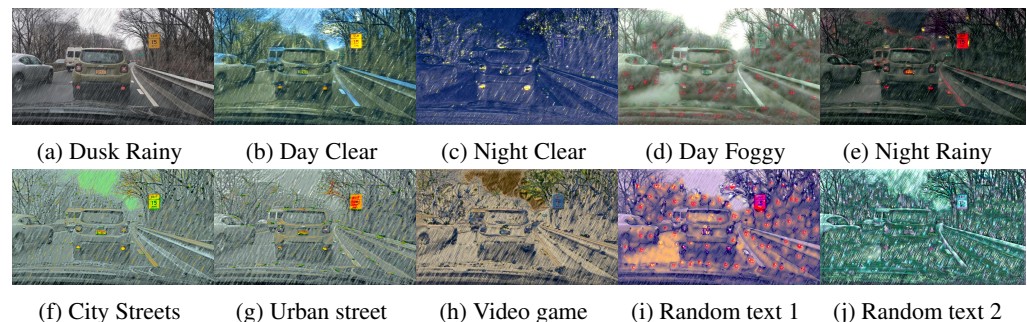

(a) Dusk Rainy     (b) Day Clear     (c) Night Clear     (d) Day Foggy     (e) Night Rainy

(f) City Streets     (g) Urban street     (h) Video game     (i) Random text 1     (j) Random text 2

Figure 6: Stylized images of different styles with an image sampled from Dusk Rainy domain of the DWD dataset. (a) is the original target image, and (b)-(j) are text-based stylized images with corresponding description in Table 21.

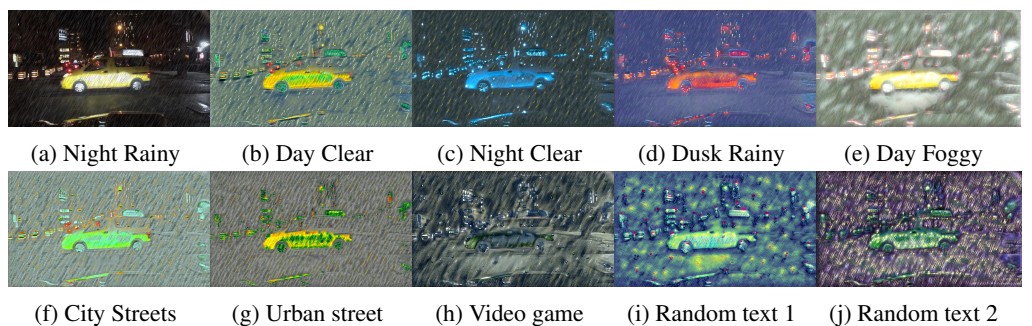

(a) Night Rainy     (b) Day Clear     (c) Night Clear     (d) Dusk Rainy     (e) Day Foggy

(f) City Streets     (g) Urban street     (h) Video game     (i) Random text 1     (j) Random text 2

Figure 7: Stylized images of different styles with an image sampled from Night Rainy domain of the DWD dataset. (a) is the original target image, and (b)-(j) are text-based stylized images with corresponding description in Table 21.

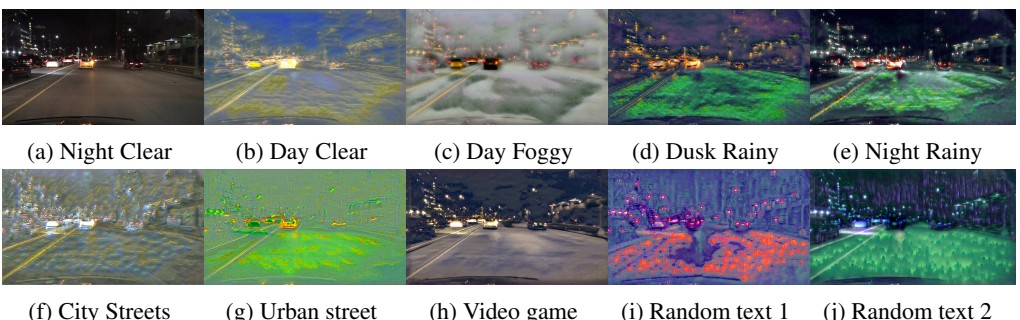

(a) Night Clear     (b) Day Clear     (c) Day Foggy     (d) Dusk Rainy     (e) Night Rainy

(f) City Streets     (g) Urban street     (h) Video game     (i) Random text 1     (j) Random text 2

Figure 8: Stylized images of different styles with an image sampled from Night Clear domain of the DWD dataset. (a) is the original target image, and (b)-(j) are text-based stylized images with corresponding description in Table 21.

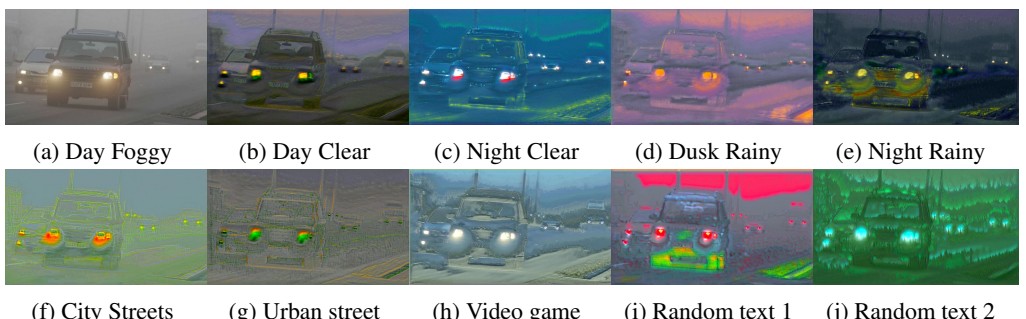

|  |  |  |  |  |
|---|---|---|---|---|
| (a) Day Foggy | (b) Day Clear | (c) Night Clear | (d) Dusk Rainy | (e) Night Rainy |
| (f) City Streets | (g) Urban street | (h) Video game | (i) Random text 1 | (j) Random text 2 |

Figure 9: Stylized images of different styles with an image sampled from Day Foggy domain of the DWD dataset. (a) is the original target image, and (b)-(j) are text-based stylized images with corresponding description in Table 21.

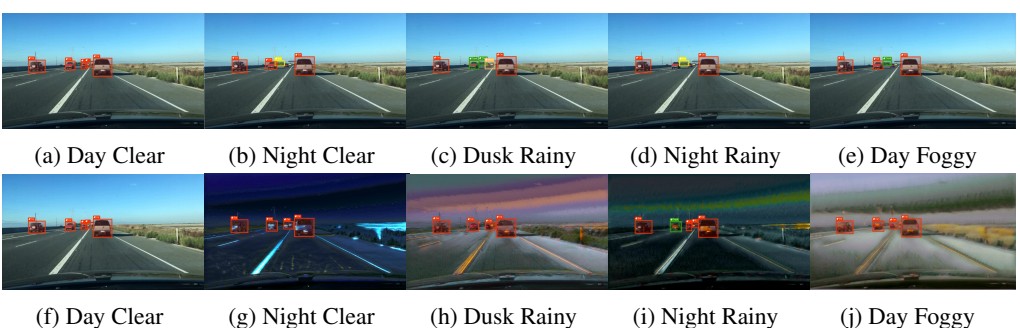

|  |  |  |  |  |
|---|---|---|---|---|
| (a) Day Clear | (b) Night Clear | (c) Dusk Rainy | (d) Night Rainy | (e) Day Foggy |
| (f) Day Clear | (g) Night Clear | (h) Dusk Rainy | (i) Night Rainy | (j) Day Foggy |

Figure 10: Prediction comparison on original and stylized images of different styles for each source model trained on Day Clear (a, f), Night Clear (b, g), Dusk Rainy (c, h), Night Rainy (d, i) and Day Foggy (e, j). (a)-(e) is the original target image, and (f)-(j) are text-based stylized images with corresponding description in Table 21. The colors mean different class labels (red: car, yellow: bus, green: truck).

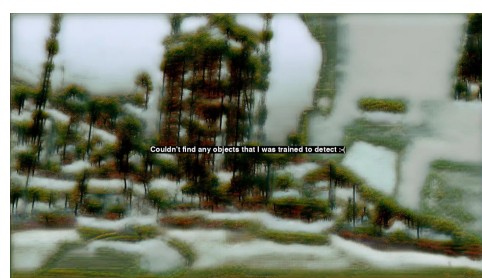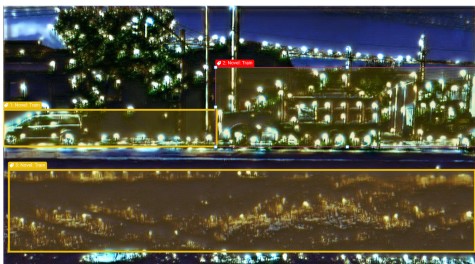

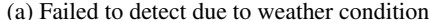

    (a) Failed to detect due to weather condition.             (b) Fail to detect due to bad lighting.

Figure 11: Example of failures.

### C.5 COMPARISON OF TFA WITH EXISTING TEXT-BASED STYLE TRANSFER METHODS

In this section, we present the augmented images generated with different text-based style transfer methods including CLIPstyler Kwon & Ye (2022), PODA Fahes et al. (2023) and our TFA using the same pre-trained CLIPstyler image decoder. In Figure 12, we present the detection results of different stylized images using the same model trained on Day Clear domain. The detection results of the two original images sampled from Day Clear domain serve as oracle results. Compared to CLIPstyler and PODA, TFA manage to maintain the details of the original images while rendering corresponding styles, such as foggy style and night style. The detection performance of TFA aligns closely with the oracle performance. While CLIPstyler misclassifies car as bus, bus as truck with Day Foggy style augmentation and fails to detection truck with Night Clear style augmentation. PODA renders a very strong style for the augmentation that even lose most of the details in the original content, which is somewhat reasonable since PODA only updates its style statistics with CLIP style loss without a content preservation regularization.

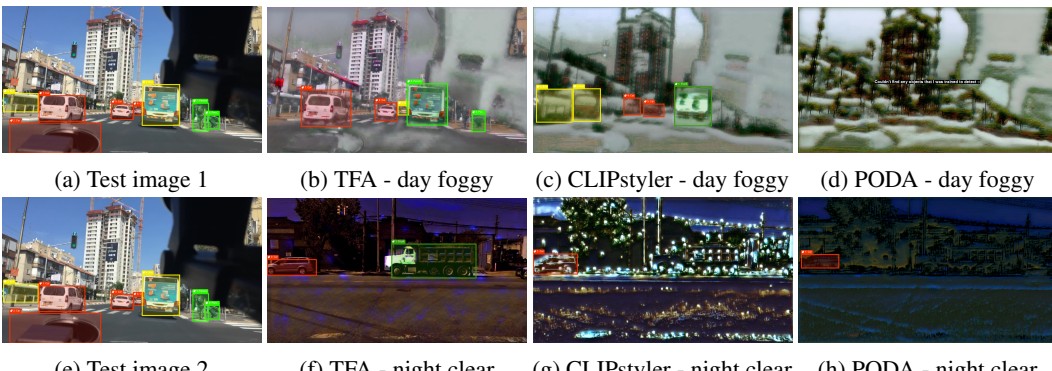

  (a) Test image 1       (b) TFA - day foggy     (c) CLIPstyler - day foggy    (d) PODA - day foggy

  (e) Test image 2       (f) TFA - night clear    (g) CLIPstyler - night clear   (h) PODA - night clear

Figure 12: Two images (a)/(e) from the Day Clear domain are stylized with different domain styles: Day Foggy and Night Clear, using different text-based style transfer methods: TFA (ours), CLIPstyler and PODA.

### C.6 COMPARISON OF DOMAIN ADAPTATION METHODS WITH ACCESS TO THE SOURCE DATA

We compare our method with *Unsupervised Domain Adaptation (UDA)* methods which have access to the source data including SW Pan et al. (2019), IBN-Net Pan et al. (2018), IterNorm Huang et al. (2019), ISW Choi et al. (2021), PODA Fahes et al. (2023), CLIP-Aug Vidit et al. (2023), and S-DGOD Wu & Deng (2022). From Table 22, we observe that even access to the source data, our model outperformes the UDA methods with access to the source data in the Night Clear and Night Rainy domain. This is attribute to the incorporation of multi-source pre-trained models. We effectively utilize the knowledge from different pre-trained source models with our proposed mean-teacher framework and benefit the generalization to the target domain without needing the access to the source data. As for the Dusk Rainy and Day Foggy domains, our method achieves comparable performance as PODA and CLIP-Aug, and outperforms the rest UDA methods, which helps demonstrate the

effectiveness of our approach. This indicates that leveraging multi-source pre-trained models with our mean-teacher framework provides a strong advantage in domain adaptation, even without direct access to the source data. Our proposed text-based augmentation successfully reduce the domain gap between the target images and the pre-trained models. Our method generalizes well across diverse target domains, outperforming traditional UDA methods in challenging conditions such as Night Clear and Night Rainy while maintaining competitive results in other domains like Dusk Rainy and Day Foggy.

Table 22: Multi-source domain adaptation results (mAP). For each target domain, Day Clear and the rest three domains are used as the source domains for the multi-source setting. For the single-source UDA and SFDA, Day Clear is used as the source following the typical setting Wu & Deng (2022); Vidit et al. (2023); Fahes et al. (2023).

| Method | Source | mAP | | | |
| --- | --- | --- | --- | --- | --- |
| | | NC | DR | NR | DF |
| SW | ✓ | 33.4 | 26.3 | 13.7 | 30.8 |
| IBN-Net | ✓ | 32.1 | 26.1 | 14.3 | 29.6 |
| IterNorm | ✓ | 29.6 | 22.8 | 12.6 | 28.4 |
| ISW | ✓ | 33.2 | 25.9 | 14.1 | 31.8 |
| S-DGOD | ✓ | 36.6 | 28.2 | 16.6 | 33.5 |
| CLIP-Aug | ✓ | 36.9 | 18.7 | 18.7 | 38.5 |
| PODA | ✓ | 43.4 | **40.2** | 19.5 | **44.4** |
| **Ours** | ✗ | **44.5** | 32.5 | **20.3** | 38.4 |

## D  POTENTIAL SOCIAL IMPACT AND LIMITATIONS

Source-data free domain adaption has the potential to significantly expand the usage of domain adaptation in more diverse settings with various constraints, such as edge devices with limited storage and applications with privacy concerns. It provides a cost-effective way to perform domain adaptation as pre-trained source models are more efficient to transfer than large datasets. It is also more memory-efficient to save the pre-trained source models versus a large training dataset. When considering data privacy, using a pre-trained source model eliminates the risk of sensitive information leaking. The proposed approach also improves existing methods based on a single source model. By simultaneously considering multiple source models, the domain gap can be effectively reduced. Nevertheless, when all the source domains exhibit a large gap as compared with the target domain, the object detection performance will naturally degrade. In this case, it is important to detect such situations and seek other potential sources for adaptation. To this end, an interesting future direction is to perform uncertainty-aware domain adaption to automatically detect the potential domain gap or choose more semantically similar domains for adaptation.

Another potential limitation is that our method requires a domain description. However, we clarify that our method only requires high-level and general information of the source domain, instead of the precise low-level details. This is much less demanding than manually labeling many data samples and some general knowledge of the domain is adequate. Furthermore, since no specific source data is required, it ensures privacy with no storage and transmission overhead. As shown in Figure 4, we have conducted an ablation of using different text descriptions of different source domains. The results demonstrate that any relevant text description can lead to reasonably good performance. Proper descriptions lead to better performance, which further reduces the requirement of the domain description, making it much more easily accessible than directly accessing source data itself. For example, for the Cityscapes dataset, we don't need to include each city names in the description, a simple yet general description such as "Driving in the city" is sufficient and able to produce superior performance. Such descriptions only require very basic understanding of the dataset and don't require much expertise.

# E    SOURCE CODE

For source code, please refer to `https://anonymous.4open.science/r/sfda_aug-ADB0/Readme.txt`.

# F    LLM USAGE STATEMENT

Large Language Models (LLMs) were used solely to aid in polishing the writing and improving the clarity of exposition. No part of the research ideation, experimental design, implementation, or analysis relied on LLMs. The authors take full responsibility for the content of this paper.

