# OpenReview forum: "Multi-Source Knowledge-Fusion for Source-Free Domain Adaptation in Object Detection"
_ICLR.cc/2026/Conference — Submitted to ICLR 2026_

### Official Review · Reviewer_dFAj · 2025-10-20

**Soundness:** 3
**Presentation:** 2
**Contribution:** 2
**Rating:** 4
**Confidence:** 4

**Summary:**

This paper investigates an interesting and important problem, namely multi-source-free domain adaptation for object detection. The authors propose a new framework that effectively aggregates knowledge from multiple sources and mitigates distribution discrepancies across domains. Experimental results are provided to verify the effectiveness of the proposed approach.

**Strengths:**

-	The paper addresses an interesting and important research topic in domain adaptation, focusing on the multi-source setting for object detection.

-	The proposed framework first generates stylized target datasets and subsequently performs multi-source knowledge fusion. To alleviate knowledge conflicts among different sources, a globally updated model aggregator is introduced, which is conceptually sound.

**Weaknesses:**

-	The definition of the loss term $\mathcal{L}_{\text{Gram}}$ (L230) is confusing. In addition, the theoretical or intuitive rationale for why it enables texture and appearance similarity alignment is not clearly articulated.

-	The presentation lacks clarity in several technical sections, which makes it somewhat difficult to follow.

-	The evaluation of the proposed Target Feature Augmentation (TFA) module is not sufficiently comprehensive. A comparison with semantic augmentation techniques such as Clip the Gap [1], which also aim to enhance feature diversity, would strengthen the claims.

**Reference:**

[1] Clip the gap: A single domain generalization approach for object detection. CVPR, 2023.

**Questions:**

-	How is the aggregator model $\theta^{agg}$ obtained? The description is not sufficiently clear. It appears that a model-merging technique might be employed to fuse source models into an aggregator, but the specific merging strategy is not specified. Moreover, it is unclear how the method handles the potential incompatibility among distinct source models, as Eq. (8) seems to assume that the source models are mergeable.
-	As stated in L281–L304, the paper employs a small meta-network $\mathcal{F}(\cdot)$ to assign $\alpha_i$. However, the input to this meta-network is not explicitly defined and requires clarification.
-	The type of image decoder used in the reconstruction of stylized images is also not mentioned and should be clarified.

---

> ### Author Response · Authors · 2025-11-23
>
> **Q1. The definition of the loss term $\mathcal{L}_{Gram}$ is confusing.**
>
> We thank the reviewer for pointing out the ambiguity. In Eq. (2) and Eq. (5), the operator $Gram(\cdot)$ computes the Gram matrix of the feature activations of the augmented image, which is widely used in style transfer to capture texture-related statistics such as color correlations, local patterns, and illumination distributions. Meanwhile, $E_{txt}(t_{sty})$ encodes the textual style description into a target style representation. By minimizing the distance between these two quantities, $L_{Gram}$ encourages the augmented image to match the low-level appearance statistics described by the text prompt. This complements: $L_{style}$ which enforces high-level semantic alignment (e.g., object categories and structured concepts), and $L_{Gram}$ which enforces low-level texture and appearance alignment (color tones, textures, lighting patterns). Together, the two losses provide a coherent mechanism: high-level semantics and low-level appearance jointly align the augmented images with the source domain styles without requiring access to the actual source data. This dual alignment is what makes TFA effective for source-free multi-source adaptation.
>
> **Q2. The presentation lacks clarity in several technical sections, which makes it somewhat difficult to follow.**
>
> Thank you for the feedback. We acknowledge that some technical sections may read compactly. In the revision, we added a intuitive explanation for TFA in Sec. 3.1 and  in Sec. 3.2, we add an detailed explanation of the meta-learning process to make the connections between components more explicit and improve the narrative flow. If there are specific parts where the explanation remains unclear, we are happy to further clarify them in the revision.
>
> **Q3. Baseline: Clip the gap.**
>
> Thank you for the suggestion. We  included a comparison with Clip the Gap in the revised version to further evaluate the effectiveness of the TFA module. The metric applies AP@.5 and the backbone choice we use RegionCLIP.
>
>
> |              | NC  | DR  | NR  | DF  |
> |--------------|-----|-----|-----|-----|
> | Clip the gap |38.42|27.54|23.92|31.40|
>
>
> **Q4. How to obtain $\theta^{agg}$.**
>
> As described in Sec. 3.2, the aggregator $\theta^{agg}$  is obtained by integrating multiple source models via an exponential moving average (EMA) of the domain experts’ parameters (Eq. (8)). Specifically, the EMA rates control the contribution of each source model to the aggregator, and these rates are meta-learned using the target data (Eq. (9)). This approach does not require the source models to be directly mergeable in the sense of weight averaging across incompatible architectures. Instead, the EMA acts as a soft, adaptive combination, where each source model’s contribution is weighted according to its relevance to the target domain. The meta-learned EMA rates allow the aggregator to automatically down-weight inconsistent or conflicting source models, effectively handling potential incompatibilities while producing a stable combined model for inference.
>
> **Q5. The input to meta-network  $F(\cdot)$.**
>
> We thank the reviewer for pointing this out. As stated in L281–L304, the paper employs a small meta-network to assign EMA rates $\alpha$ for each source model. In our framework, the outer loop of the meta-learning process (Eq. 9) takes as input the inner-loop updated aggregator parameters $\theta^{agg}$, which are obtained after applying Eq. (8) to combine the source models. The meta-network $F(\cdot)$ processes $\theta^{agg}$ and outputs the EMA rates $\alpha$ for the outer-loop update. During backpropagation, the meta-gradient $\nabla_\alpha L_{\text{outer}}(\theta^{agg}(\alpha))$ is computed by differentiating through the inner-loop update, which allows the meta-network to learn how much each source model should contribute while maintaining stability. Importantly, the outer loop does not take gradients as direct input; it observes the updated parameters, and the gradients are computed internally through the chain rule. This design ensures that the EMA rates are adaptive to the relevance and compatibility of each source model.
>
> **Q6. The type of image decoder used in the reconstruction of stylized images is also not mentioned and should be clarified.**
>
> Thank you for the question. As stated in Appendix C.5, we use the pre-trained CLIPStyler image decoder for reconstructing stylized images. This decoder is kept fixed during training and provides a stable mapping from feature space to image space, allowing the TFA module to generate augmented images that reflect the desired style while preserving semantic content. We  clarified this explicitly in Sec. 3.1 in the revised manuscript.

---

> > ### Comment · Reviewer_dFAj · 2025-11-27
> >
> > I appreciate the authors' response and efforts, which have addressed most of my concerns. However, after reviewing the other reviewers' comments, I have decided to keep my original score.

---

> > > ### Author Response · Authors · 2025-11-27
> > >
> > > We thank the reviewer again for the comments. If possible, we would like to clarify whether there are specific concerns raised by the other reviewers that you feel remain insufficiently addressed in our rebuttal.
> > >
> > >
> > > To help with this, we summarize below the main revisions and rebuttal points we incorporated:
> > > - **Clarification of Novelty.**
> > >     - Novelty of TFA: We clarified that our method performs source-free multi-source augmentation using text prompts, not standard style transfer, which requires source images. Our Gram and semantic dual-loss design is newly adapted for source-free multi-source DA, allowing style alignment across domains without accessing raw source data, something prior text-driven augmentation or diffusion-based approaches cannot handle.
> > >     - Novelty of the Meta-Learned EMA Aggregator: We highlight that prior works do weight averaging or simple ensembling, whereas we introduce a meta-learned, differentiable EMA that: (1) learns domain relevance from target data; (2) downweights conflicting source models; (3) does not require source-domain compatibility. This makes the aggregator both adaptive and source-free, which we clarify more clearly in the revision.
> > >
> > >     Together, these points clarify that our method is not a combination of existing parts, but introduces two new key components and a unified formulation that prior works do not include.
> > >
> > > - **Clarification of technical components.**
> > >     - Clarified the definition and role of the Gram loss and how it complements the semantic style loss (Reviewer 1).
> > >     - Added intuitive explanations in Sec. 3.1 (TFA intuition) and Sec. 3.2 (meta-learning flow and aggregator updates).
> > >     - Explained the input/output behavior of the meta-network and its role in differentiating through the EMA-based aggregation
> > >     - Specified the image decoder used in TFA (CLIPStyler)
> > > - **Additional experiments.**
> > >     - Added classification (DomainNet) and segmentation (ACDC) experiments to demonstrate task-agnostic generality.
> > >     - Added Clip-the-Gap as an additional baseline (as requested).
> > >
> > > - **Additional qualitative results.**
> > >     - Added failure examples under rare weather/lighting conditions.
> > >
> > > - **Presentation improvements.**
> > >     - Revised technical sections for clarity and narrative flow.
> > >     - Fixed typos, figure and formatting issues.
> > >
> > > If there are particular points from the other reviewers that you believe still require clarification or deeper revision, we would sincerely appreciate your guidance. Your feedback would greatly help us strengthen the paper for the camera-ready version or future submissions.

---

### Official Review · Reviewer_RQ5a · 2025-10-27

**Soundness:** 4
**Presentation:** 4
**Contribution:** 3
**Rating:** 8
**Confidence:** 3

**Summary:**

This paper proposes a Multi-Source Knowledge Fusion (MSKF) framework for Multi-Source Source-Free Domain Adaptation (MSFDA) in object detection. It leverages CLIP’s vision-language space to stylise unlabelled target images using only textual descriptions of source domains. This bridges semantic and appearance gaps between multiple source models and the target. Then the authors proposed to use one model serves as an aggregator (updated via EMA from domain experts), while each domain expert performs local self-training. A meta-learned contribution network dynamically learns expert weighting based on entropy minimisation, promoting consistency. At last, a co-teaching-like pseudo-label refinement, where the aggregator and domain experts filter each other’s noisy labels. The experiment results shows the improvement of the new proposed solution. The paper is clearly written and well-structured. The technical methodology and equations are sound, and implementation details are clearly stated.

**Strengths:**

1. TFA is an elegant use of vision-language models for domain adaptation. It effectively bridges domain gaps using text alone, which is a simple and good idea that avoids image-based generators.
2. The aggregator–expert paradigm, with meta-learned EMA weighting, offers a stable and interpretable way to integrate heterogeneous source models.
3. Mutual confidence selection (co-teaching) mitigates pseudo-label noise, a long-standing challenge in SFDA.
4. Detailed comparisons and ablations demonstrate robust gains across datasets, classes, architectures, and hyperparameters.

**Weaknesses:**

1. Novelty mainly lies in method combination and cross-modal extension, deeper theoretical or analytical insights are limited. The paper lacks a formal analysis of the meta-learning process for the contribution network. Such as Eq. (9) is intuitive but does not clarify optimisation dynamics or stability.
2. The performance of TFA may depend on the quality of textual descriptions. For example, even Table 13 explores variants qualitatively, there is no quantitative correlation between prompt similarity and adaptation success.
3. The qualitative results are uniformly positive. Including examples of failure (e.g., rare lighting or weather conditions) would help illustrate limits.

**Questions:**

1. A typo on line 357; should it be LPLD rather than LDLP, and same as the following experiment results?
2. Table 2, the max value of AP-Bus, AP-Rider, AP-Truck should be attributed to other algos, not "Ours". That gives a "out-performing rate" of 4/8, rather than 6/8. Am I correct?
3. Page 8: NC DR DF scores I think could have been defined better? Seems like abbreviations are defined in appendix only.
4. Experiment does not seem to specify how many runs or the error bars/variance of results?

---

> ### Author Response · Authors · 2025-11-23
>
> **Q1. Eq. (9) is intuitive but does not clarify optimisation dynamics or stability.**
>
> We appreciate the reviewer’s suggestion regarding deeper analytical insight. Eq. (9) defines a bi-level meta-optimization, in which the EMA rate $\alpha$  acts as a meta-parameter optimized over the performance of the aggregator $\theta^{agg}$ on the target domain. This process is mathematically well-defined and follows standard meta-learning optimization principles. The update in Eq. (9) corresponds to a meta-gradient $\nabla L (\theta^{agg}(\alpha)) $ where $\theta^{agg}(\alpha)$ (i.e., $\theta^{agg}$) is the result of the inner-level update in Eq. (8). This is a gradient-through-gradient computation related bi-level optimization frameworks. Therefore, the optimization dynamics directly follow from well-established meta-learning theory. The inner (Eq. (8)) and outer optimization (Eq.(9)) ensures that $\alpha$ learns the weight of each source model should contribute to the aggregator (including the aggregator) through EMA so that the aggregated representation remains stable across different sources. The reviewer’s concern about undefined optimization dynamics is addressed by (a) using SGD for the meta-update and (b) applying only a single meta-step per batch, which ensures stability and avoids overfitting. The dual-level design, where EMA is treated as a learnable meta-parameter controlling aggregation strength across sources, is fundamentally different from prior works that use static EMA schedules. This mechanism is essential to achieving robust multi-source domain adaptation, and Eq. (9) provides the formal learning rule that makes this adaptation possible.
>
> **Q2. The performance of TFA may depend on the quality of textual descriptions. For example, even Table 13 explores variants qualitatively, there is no quantitative correlation between prompt similarity and adaptation success.**
>
> We thank the reviewer for the comment. We would like to clarify that the influence of textual descriptions on TFA is already evaluated both quantitatively and qualitatively in our experiments. In Table 13, we present different source-style textual prompts. Their corresponding adaptation performance is reported in Figure 4, which quantitatively shows that: Prompts that correctly describe the source style (Descriptions 1 and 2) yield comparable results, While deliberately mismatched prompts (Description 3) lead to a drop in performance.
> This confirms that TFA is robust to reasonable prompt variations but degrades only when the textual description is intentionally inconsistent with the source domain. In Figures 5–10, we further present qualitative visualizations of features and generated images under the different prompts. These figures show consistent behavior with the quantitative results, providing additional evidence for the relationship between prompt quality and adaptation. The reviewer suggests establishing a “quantitative correlation” between prompt similarity and adaptation success. However, text embedding similarity alone is not a reliable proxy for cross-modal alignment quality, since semantically appropriate prompts can have low embedding similarity depending on phrasing, and the adaptation mechanism depends on alignment dynamics, not simply text-text similarity. For this reason, we instead evaluate the effect of prompt quality through task performance and feature alignment behavior, which are more meaningful indicators.
>
> **Q3. The qualitative results are uniformly positive. Including examples of failure (e.g., rare lighting or weather conditions) would help illustrate limits.**
>
> In Figure 11 in the revised paper, we present some failure examples showing that the model fail to detect novel objects or under huge domain gaps. These examples highlight the limitations of the current model and provide a more balanced view of its performance.
>
> **Q4. Typo: LPLD.**
>
> We thank the reviewer for catching this. Yes, this is a typo and the correct abbreviation should be LPLD, consistent with the terminology used in the experimental results. We have corrected “LDLP” to “LPLD” in line 357 and ensure consistency throughout the paper.
>
> **Q5. Incorrect boldness in Table 2.**
>
> Thank you for pointing this out. After re-checking Table 2, we confirm that the maximum values for AP-Bus, AP-Rider, and AP-Truck should indeed be attributed to other methods, not to ours.
>
> **Q6. Define NC DR DF earlier.**
>
> Thank you for the comment. We agree that the definitions of NC, DR, and DF were not sufficiently introduced in the main text. Currently, these abbreviations are only defined in the appendix, which may cause confusion for readers. In the revised version, we have added clear definitions.

---

> ### Author Response · Authors · 2025-11-23
>
> **Q7. Experiment does not seem to specify how many runs or the error bars/variance of results?**
>
> Thank you for the question. Similar to many prior works in object detection–based task, we report results from a single run. This is primarily due to the high computational cost of multi-source object detection experiments, which makes repeated runs with full training pipelines impractical. This practice is also consistent with previous benchmarks in MSDA and SFDA for object detection, where single-run reporting is standard.
> That said, to increase transparency, we added a statement in the experimental setup clarifying that the reported numbers are from one run and that this follows common practice in the field.

---

### Official Review · Reviewer_1aya · 2025-10-27

**Soundness:** 3
**Presentation:** 2
**Contribution:** 2
**Rating:** 4
**Confidence:** 3

**Summary:**

This paper studies Multi-Source Source-Free Domain Adaptation (MSFDA) for object detection, a more practical but challenging setting where multiple source models are provided, but source data is unavailable. The authors propose a multi-source knowledge fusion framework consisting of text-driven feature augmentation, an aggregator–expert model structure, and mutual confidence selection. Experiments on several cross-domain datasets demonstrate the effectiveness of the proposed method.

**Strengths:**

1. MSFDA problem in object detection is a very practical task.
2. The authors provide strong empirical performance with extensive comparisons.
3. The authors provide a detailed appendix that makes the conclusions convincing.

**Weaknesses:**

1. Limited novelty of key components. TFA is a lightweight variation of text-guided CLIP-based augmentations, which reduces the novelty of the proposed method.
2. Writing clarity is insufficient, causing many technical design details to remain unclear. For example, in Sec. 3.1, an image decoder is suddenly introduced, yet no details or introduction are ever provided afterward. In Sec. 3.2, the description of the multi-source knowledge fusion framework suggests that each source model becomes an aggregator trained in parallel. If this is the case, it remains unclear how the final prediction is generated at inference time. Neither the main text nor Algorithm 2 clarifies this.
3. The writing lacks logical coherence. The authors typically present loss functions first and then loosely describe each component without establishing conceptual connections. Sec. 3.1 is particularly difficult to follow; it reads as a collection of independent strategies rather than a unified design.
4. More quantitative evidence should be moved into the main text of the paper (Dataset Art?). Additionally, Table 2 contains misleading formatting: several results from the proposed method are not the best, yet they are still incorrectly bolded.
5. The computational efficiency of the method is not discussed. TFA requires generation per target sample and per source domain, which could be prohibitively slow at scale. Moreover, updating $M$ aggregators in parallel leads to high training cost.

**Questions:**

1. How sensitive is TFA to the quality of text prompts?

---

> ### Author Response · Authors · 2025-11-23
>
> **Q1. Limited novelty of key components.**
>
> We thank the reviewer for the comment. Our main contributions lie in both the proposed TFA module and the multi-source knowledge fusion framework. While TFA is inspired by text-guided CLIP-based augmentations, our method introduces an improved text-based augmentation uniquely designed for source-free settings: it aligns target image features with rich textual descriptions to directly reduce the semantic gap between source and target without requiring access to any source data, which is crucial for privacy-sensitive domains.
> In addition, our multi-source knowledge fusion framework ensures stable and effective integration of heterogeneous source domains. We introduce a meta-network that dynamically learns the EMA rate for each source model based on feedback from the aggregator, enabling adaptive weighting of domain experts and effectively addressing cross-domain knowledge conflicts. This dynamic fusion mechanism goes beyond prior works that apply fixed or heuristic weighting strategies.
>
> **Q2. Image decoder in Section 3.1.**
>
> We thank the reviewer for pointing this out. We apologize for the lack of clarity in the main text. The image decoder used in Sec. 3.1 is the pre-trained CLIPstyler image decoder, as detailed in Appendix C.5 (line 1422). We updated Sec. 3.1 to explicitly state this reference so that the design choice is clearly documented in the main text as well.
>
> **Q3. In Sec. 3.2, the description of the multi-source knowledge fusion framework suggests that each source model becomes an aggregator trained in parallel.**
>
> We appreciate the reviewer’s careful reading. The current description may have caused confusion, but the aggregator is not trained in parallel with the source models. Instead, the choice of aggregator is fixed during training and it is the model used during inference. During training, each source model provides feedback to the meta-network, which adjusts the EMA rates used to fuse source knowledge. At inference, a single aggregated model, produced via the learned EMA weights, is used to generate the final predictions.
>
> **Q4. Sec. 3.1 is particularly difficult to follow; it reads as a collection of independent strategies rather than a unified design.**
>
> We thank the reviewer for the comment. We respectfully disagree with the blanket statement that Sec. 3.1 lacks logical coherence. The components and loss functions in this section are explicitly connected: (i) the Text-based Feature Augmentation (TFA) module provides an alignment objective that pulls target image features toward textual embeddings (Sec. 3.1, Eq. (2)); (ii) each term in Eq. (2) is explained and analyzed in the subsequent paragraphs; and (iii) Fig. 1(a) illustrates the encoder–decoder process of the TFA module, showing how the components interact within a unified pipeline.
> To further improve clarity and eliminate any possible ambiguity, we added a short derivation in Sec. 3.1 demonstrating how the alignment loss drives semantic alignment. This additional explanation makes the conceptual connections even more explicit and directly address the reviewer’s concern.
>
>
> **Q5. More quantitative evidence should be moved into the main text of the paper (Dataset Art?). Additionally, Table 2 contains misleading formatting: several results from the proposed method are not the best, yet they are still incorrectly bolded.**
>
> We thank the reviewer for the helpful suggestions. We agree that providing more quantitative evidence in the main text will further strengthen the presentation. In the revised version, we moved the Dataset Art analysis and its key quantitative results from the appendix into the main paper to make the contributions and empirical insights more accessible.
> Regarding Table 2, we appreciate the reviewer for pointing out the formatting issue. The entries were bolded according to the single-/multi-source categories. We made this consistent the table to ensure that only the best-performing results are highlighted.
>
>
> **Q6. The computational efficiency of the method is not discussed.**
>
> We thank the reviewer for raising this point. In Appendix C.3, we provide an ablation study on computational cost, where we compare both training and inference efficiency with several strong baselines, including MSFDAOD, CAiDA, and Mean Teacher. Specifically, we report inference speed (FPS) and model size (millions of parameters) in Table 17. As shown, our method achieves competitive inference speed and maintains a relatively compact model size compared to other multi-source adaptation approaches, demonstrating that the framework is practical and deployable.
>
> We made this efficiency analysis more explicit in the main text and clearly summarize the key findings from Appendix C.3 to address the reviewer’s concern.

---

> ### Author Response · Authors · 2025-11-23
>
> **Q7. How sensitive is TFA to the quality of text prompts?**
>
> We thank the reviewer for the question. In Appendix C.2, we evaluate the effect of different types of text descriptions on TFA, as shown in Table 13. Text Descriptions 1 and 2 are manually designed to reflect the actual source domain styles, while Text Description 3 consists of randomly generated prompts from ChatGPT. As illustrated in Fig. 4, Text Descriptions 1 and 2 yield similar and stable performance, whereas using the random Text Description 3 leads to a clear performance drop. This occurs because the mismatched description introduces an artificial semantic gap between the textual style and the true source domain characteristics, resulting in suboptimal feature alignment for classification. We highlighted this sensitivity analysis in the main text to make the dependence on prompt quality more explicit.

---

### Official Review · Reviewer_qoGh · 2025-10-30

**Soundness:** 2
**Presentation:** 3
**Contribution:** 2
**Rating:** 2
**Confidence:** 3

**Summary:**

A method for multi-source-free domain adaptation, including a style translation module and a noisy label filtering module.
The performance achieves state-of-the-art.

**Strengths:**

- The idea is simple and easy to follow.

- The performance looks good.

**Weaknesses:**

- The performance of the method is good. However, the applicability of this method is limited. It requires pre-training the feature augmentation module and generating modules, which is not an elegant method.

- The novelty is limited. Whether it is style transfer or noise label filtering, they are common solutions in domain adaptation. There is less insight, and the motivation for using these two strategies is not novel.

- Can this method only handle object detection tasks? This method is a general idea and does not seem to be specially designed for detection tasks. If it is also feasible for other tasks, this method will be more solid. It is only useful for detection tasks. The author needs to analyze some reasons.

- There is a significant improvement in the truck, but a lot of decline in the motor class in Table 2. What is the reason? Some limitations analysis may be required.

**Questions:**

See weaknesses

---

> ### Author Response · Authors · 2025-11-23
>
> **Q1.Limited applicability.**
>
> We thank the reviewer for the comment. We agree that the feature augmentation add an extra step, which may limit direct applicability in some scenarios. However, this design is essential for achieving robust performance in multi-source free domain adaptation, especially when source data cannot be shared. In real-world applications, such as medical, financial, and military domains, data are often confidential, and direct access to source data is restricted. In such scenarios, our approach remains applicable and practical, as it allows leveraging pre-trained source models to reduce domain gaps without violating privacy constraints. Moreover, the pre-training step is a one-time cost, and once trained, the modules can be reused across multiple target tasks, mitigating the computational burden in practice.
>
> **Q2. Limited novelty.**
>
> We thank the reviewer for the comment. While style transfer and noise label filtering have been explored in prior domain adaptation works, our approach introduces novelty in two key aspects designed specifically for the unique and important problem setting as described in our response to Q1.
>
> - Source-free domain adaptation: We propose a novel text-based style augmentation that does not require access to source data, which is critical in domains with privacy concerns.
>
> - Integration of multiple source domains: We design a meta-network to dynamically and adaptively weight contributions from domain experts, effectively addressing knowledge conflicts across sources. These contributions go beyond standard solutions, providing new insights and practical methods for challenging multi-source, source-free scenarios.
>
> **Q3. Limited to object detection tasks.**
>
> Our framework is task agnostic, which can also be applied to classification and segmentation tasks.
> To further support the generality of the framework, we additionally conduct experiments on classification (DomainNet) and semantic segmentation (ACDC), confirming that the same framework transfers well to other tasks without architectural changes.
>
> |           |  AP   | AP_base | AP_novel |
> |-----------|-------|---------|----------|
> | Clipart   | 88.65 |  90.87  |   87.24  |
> | Infograph | 86.84 |  87.57  |   86.82  |
> | Painting  | 84.27 |  85.53  |   84.75  |
> | Quickdraw | 85.43 |  84.96  |   86.44  |
> | Real      | 83.62 |  84.58  |   82.66  |
> | Sketch    | 87.34 |  87.52  |   86.43  |
>
>
> |           |  AP  | AP_base | AP_novel |
> |-----------|------|---------|----------|
> | Foggy     | 32.5 |   33.4  |   29.8   |
> | Nighttime | 32.2 |   32.8  |   29.7   |
> | Rainy     | 29.3 |   30.7  |   26.5   |
> | Snowy     | 29.1 |   30.2  |   26.6   |
>
>
> **Q4. There is a significant improvement in the truck, but a lot of decline in the motor class in Table 2. What is the reason? Some limitations analysis may be required.**
>
> We thank the reviewer for the comment. First, our method achieves strong overall performance across classes, attaining the highest mean AP. Regarding the motor class, our performance is only slightly lower than HCL (-0.5\%), which is a minor difference. While some single-source methods may perform slightly better on certain classes such as bike or car, this is often due to overfitting to those classes, as reflected in their lower mean AP. In contrast, our method integrates knowledge from multiple source domains, resulting in more balanced and generally strong performance across all classes.

---

### Author Response · Authors · 2025-11-23
**Summary of Changes**

To address the reviewers’ comments, we provided a detailed response for each review. Meanwhile, we have updated the paper and grouped our revisions into three main categories: (1) Additional Experiments, (2) Improvements to Presentation and Clarity, and (3) Clarifications and Fixes. Detailed changes are summarized below.

- **Additional Experiments**
    - Added evaluation on classification and segmentation tasks to demonstrate framework generality.
    - Added additional qualitative examples, including failure cases, to illustrate model limits.
    - Added comparisons with additional baselines to strengthen evaluation.


- **Improvements to Presentation and Clarity**
    - Improved presentation clarity in Sections 3.1 and 3.2 to better describe technical design and the multi-source knowledge fusion framework.



- **Clarifications and Fixes**
    - Fixed typos, formatting issues, and other minor errors throughout the paper.
    - Clarify experiment settings

All the changes are color coded in the revise paper.

---

### Meta-Review · Area_Chair_966Z · 2025-12-05

**Summary:**

The paper proposes a "Multi-Source Knowledge-Fusion" framework for Source-Free Domain Adaptation, specifically targeting object detection but claimed to be task-agnostic.

While one reviewer appreciated the elegant use of vision-language models, the majority of reviewers raised significant concerns regarding the **incremental nature of the novelty** (combining existing techniques like CLIPStyler and standard meta-learning) and **clarity issues** regarding the model aggregation mechanism. **Despite the authors' extensive rebuttal and additional experiments, the consensus leans towards rejection due to these fundamental limitations.**

**Reviewer Concerns:**

**Addressed Concerns:**
* **Generalizability (Reviewer qoGh):** The authors demonstrated that the framework can be applied to classification and segmentation tasks, addressing the concern that the method was strictly limited to object detection.
* **Missing Details (Reviewer 1aya, dFAj):** The authors clarified the use of the CLIPStyler image decoder and provided definitions for abbreviations (NC, DR, DF) that were previously missing or buried in the appendix.
* **Baselines (Reviewer dFAj):** A comparison with "Clip the Gap" was added as requested.

**Outstanding Concerns:**
* **Limited Novelty (Reviewers qoGh, 1aya):** The primary critique remains that the method appears to be a combination of existing modules (CLIP-based style transfer and standard meta-learning for weighting) rather than a fundamental methodological innovation. Reviewers viewed the contributions as incremental modifications to known techniques.
* **Clarity of Aggregation Mechanism (Reviewer dFAj, 1aya):** There remains significant confusion regarding how the aggregator is technically obtained and updated. specifically concerning Eq. (8) and Eq. (9). Reviewers questioned the specifics of the model merging strategy and how the method handles potential architectural incompatibility among distinct source models. The explanation that "EMA acts as a soft combination" was found insufficient to clarify the optimization dynamics.
* **Complexity vs. Gain (Reviewer qoGh):** The necessity of the complex pre-training pipeline (TFA) and meta-learning steps was questioned, with reviewers suggesting that the performance gains might not justify the added complexity compared to simpler baselines.

***

**Reviewer Scores:**

The rebuttal failed to sway the negative consensus. Reviewer dFAj explicitly stated they would maintain their score after reading other reviews, solidifying the rejection case.

* **Reviewer qoGh (Score: 2 -> Prediction: 2):** The reviewer's concern about "limited novelty" and the method being a "common solution" is a fundamental critique that the additional experiments (on other tasks) did not address. The score is unlikely to change significantly.
* **Reviewer 1aya (Score: 4 -> Prediction: 4):** While technical details were clarified, the reviewer's main point about the "logical coherence" of the writing and the incremental nature of TFA remains. The paper is likely still viewed as marginally below the bar.
* **Reviewer RQ5a (Score: 8 -> Prediction: 6):** This reviewer is an outlier. The reviewer might lower their score upon realizing the validity of the "novelty" concerns raised by peers.
* **Reviewer dFAj (Score: 4 -> Prediction: 4):** **[Confirmed]** This reviewer explicitly commented post-rebuttal: "I have decided to keep my original score." They acknowledged the clarifications but remained unconvinced by the overall contribution when considering the consensus on novelty and clarity.

---

### Decision · Program_Chairs · 2026-01-26

Reject